# Coaxially printed magnetic mechanical electrical hybrid structures with actuation and sensing functionalities

Yuanxi Zhang [1,4], Chengfeng Pan[2,4], Pengfei Liu [1,4], Lelun Peng[1], Zhouming Liu[1], Yuanyuan Li[1], Qingyuan Wang[1], Tong Wu[1], Zhe Li [1] ✉, Carmel Majidi [3] ✉ & Lelun Jiang [1] ✉

Soft electromagnetic devices have great potential in soft robotics and biomedical applications. However, existing soft-magneto-electrical devices would have limited hybrid functions and suffer from damaging stress concentrations, delamination or material leakage. Here, we report a hybrid magnetic-mechanical-electrical (MME) core-sheath fiber to overcome these challenges. Assisted by the coaxial printing method, the MME fiber can be printed into complex 2D/3D MME structures with integrated magnetoactive and conductive properties, further enabling hybrid functions including programmable magnetization, somatosensory, and magnetic actuation along with simultaneous wireless energy transfer. To demonstrate the great potential of MME devices, precise and minimally invasive electro-ablation was performed with a flexible MME catheter with magnetic control, hybrid actuation-sensing was performed by a durable somatosensory MME gripper, and hybrid wireless energy transmission and magnetic actuation were demonstrated by an untethered soft MME robot. Our work thus provides a material design strategy for soft electromagnetic devices with unexplored hybrid functions.

Soft electromagnetic devices combine magnetic and electrical functionality with intrinsic mechanical compliance to achieve robust elasticity along with remote actuation, sensing, and energy harvesting[1,2]. These emerging classes of material systems have the potential for transformative impact in a variety of application domains, including soft robotics[3,4], wearable computing[5–8], and biomedical devices[9,10].

Current approaches to creating soft electromagnetic devices are based on heterogeneous combinations of magnetoactive elastomer composites and conductive materials. Magnetoactive materials are usually made of soft elastomers embedded with ferromagnetic particles that are magnetized with a programmed magnetization profile[11,12]. Conductive materials used in soft electromagnetic devices include

liquid metal[13–21], serpentine copper/gold patterns[4,22–27], or conductive composites consisting of a soft polymer matrix and a conductive dispersion phase, e.g. metallic particles (e.g., liquid metal[3,28–32], silver[33], gold[34]), carbon allotropes[35,36], or soft electrically conductive composites (e.g., polyaniline[37], polypyrrole[38], PEDOT:PSS[39]). These materials are typically combined using one of the following three general strategies (see Supplementary Table 1 for details). The first approach involves mixing rigid conductive filler particles within an uncured magnetoactive matrix[3,31]. Such an approach requires high concentrations of conductive filler for enhanced conductivity, but this can degrade the mechanical properties of the composite. The second and third approaches rely on interfacing magnetoactive materials with

[1]Guangdong Provincial Key Laboratory of Sensor Technology and Biomedical Instrument, School of Biomedical Engineering, Shenzhen Campus of Sun Yat-sen University, Shenzhen 518107, PR China. [2]The State Key Laboratory of Fluid Power and Mechatronic Systems, College of Mechanical Engineering, Zhejiang University, Hangzhou, Zhejiang 310027, China. [3]Soft Machines Lab, Mechanical Engineering, Carnegie Mellon University, Pittsburgh, PA 15213, USA. [4]These authors contributed equally: Yuanxi Zhang, Chengfeng Pan, Pengfei Liu. ✉e-mail: lizhe28@mail.sysu.edu.cn; cmajidi@andrew.cmu.edu; jianglel@mail.sysu.edu.cn

deterministically patterned metal conductors that achieve through geometry (e.g., serpentine, wrinkle, and wavy structures)[4,24–26,40] or with rigid electromagnetic coils[5,23,27]. As with the first and second approaches, these latter methods require the interfacing of soft and stiffness materials, which can lead to high internal stress concentrations and issues with interfacial adhesion and delamination.

A promising alternative to these methods is to use liquid phase metal alloys like eutectic gallium-indium (EGaIn) as the conductive material for supporting electrical current and electromagnetic induction. Previous efforts have explored the fabrication of core-sheath fibers consisting of the liquid metal core and a polymer sheath[41–43], including template molding and injecting[44–49], 3D shape programming and dip-coating[50,51], coaxial wet-spinning[28,52,53] and coaxial printing[42,54,55], offering the possibility to fabricate soft electromagnetic devices with complex structures (complex pattern[13,15] and multilayer structure[14,50,54]; see Supplementary Table 2 for details). However, these core-sheath fibers only have hybrid mechanical-electrical properties without magnetoactive characteristics, and soft electromagnetic devices built upon these materials lack the hybrid magnetic actuation and energy transfer functions. Although liquid metal ferrofluids[56,57] could be infused into hollow fibers for magnetically actuated deformation[58,59] while ensuring good flexibility and high electrical conductivity, these liquid metal ferrofluid-based soft electromagnetic devices often suffer from weak remanence or non-programmable magnetization[60–64], limiting their capabilities for complex shape deformation and somatosensory actuation (see Supplementary Fig. 1 and Fig. 2). Therefore, it is still challenging to develop soft electromagnetic devices with hybrid magnetic actuation, energy transfer, and somatosensory actuation functions.

In this work, we overcome these challenges by introducing a hybrid magnetic-mechanical-electrical (MME) core-sheath fiber that is patterned using a one-step coaxial printing method. The MME fiber is composed of a liquid metal core and a soft magnetoactive composite sheath that together enable programmable magnetic responsiveness, high electrical conductivity ($2.07 \times 10^6$ S/m), robust mechanical properties (150% strain limit and 0.87 MPa Young's modulus), and functional durability (remain highly conductive in tension and bending and functions in harsh environments) in a single structure. With the assistance of digitally-controlled coaxial printing, complex 2D/3D coil-structured MME devices with hybrid magnetoactive and electrically conductive characteristics can be fabricated in one step. Moreover, such printed structures are capable of combined somatosensory actuation and energy harvesting through the coupling of the magnetoactive sheath and strain-sensitive inductance enabled by the electrically conductive liquid metal core. To demonstrate the potential of the MME-based soft electromagnetic architectures, a 1D MME fiber was used as a catheter-style soft surgical tool to perform a minimally invasive electro-ablation surgery, a task that requires precise navigation and high electrical power delivery. We also developed somatosensory grippers with 2D MME structures to identify and sort objects. Lastly, a soft robot made by 2D MME is introduced to demonstrate the functionality of magnetic actuation and energy harvesting.

## Results

### Coaxial printing of the hybrid MME structures

Figure 1a illustrates the design concept of the hybrid magnetic-mechanical-electrical (MME) structure. The MME is designed with a core-sheath structure consisting of two functional components (Fig. 1a), including a soft magnetoactive composite sheath (NdFeB @ PDMS composite) and a flexible, electrically conductive core (liquid metal). In order to fabricate the hybrid MME structure with complex geometries, a coaxial printing method has been developed for integrating the functional components, as illustrated in Fig. 1b (see Supplementary Fig. 3 for the customized coaxial printer; see Supplementary Movie 1 for the process of coaxial printing the MME

structure). Briefly, a printable composite ink for the sheath was prepared by dispersing non-magnetized NdFeB microparticles (size: 5 μm) in the PDMS matrix (SE 1700); after systematic investigation of the magnetic, rheological, and mechanical properties, the composite ink was optimized to have a NdFeB to PDMS weight ratio of 1:1 (Supplementary Fig. 4). Liquid metal (EGaIn, Ga: 75 wt%, In: 25 wt%) is used as the electrically conductive core material due to its low melting temperature (about 16 °C). The composite ink and the liquid metal are fed to the outer and inner channel of a coaxial nozzle, respectively, with optimized printing parameters to fabricate the pre-cure MME structure with PDMS @ NdFeB composite as the flexible sheath and liquid metal as the core, respectively (Fig. 1b and Supplementary Movie 1). The printed pre-cure MME structure is placed in a thermal oven at 70 °C for 1 h and then magnetized in a pulsed magnetic field $\mathbf{B}_{mag}$ (about 3 T) to generate a desired magnetization profile $\mathbf{m}$ within the MME structure (see Supplementary Fig. 5 for magnetization).

After thermal curing and magnetization, the MME structure can be actuated in a controllable manner by adjusting the external actuation magnetic field $\mathbf{B}_{act}$ (Fig. 1a and Supplementary Fig. 6). Compared with existing 3D/4D printing methods reported in the literature[60,64], the coaxial printing method can integrate multi-material properties (magnetic, mechanical and electrical) in a single structure, enabling magnetically actuated deformation, deformation sensation and even wireless energy transmission/harvesting. Figure 1c exhibits the actuation of a 1D MME fiber (length: 85 mm; diameter: 830 μm) with magnetic control (Supplementary Fig. 7a). Controlled by the external magnetic field, the MME fiber can be precisely actuated to electrically connect and light LED pixels in a confined space through the liquid metal core (Fig. 1c, Supplementary Fig. 7b and Movie 2).

Further, with numerically controlled coaxial printing, 1D MME fiber can be constructed into a complex 2D MME coil structure (Fig. 1b, Supplementary Movie 1 and 3), which can be mechanically deformed by the external magnetic field. By altering the dimensions of the liquid metal core, the mechanical deformation will induce a dynamic change of inductance $\Delta L$ in the 2D MME coil, which can be exploited to sense the deformation of the MME structure. Figure 1d and Supplementary Figure 8 show a coaxially printed butterfly robot with a 2D MME coil structure as the skeleton (size: $42 \times 50 \times 0.8$ mm³), which has been designed with a customized magnetization profile $\mathbf{m}$ (Supplementary Fig. 9). The flapping angle $\alpha$ of the butterfly robot wings can be controlled by adjusting the external actuation magnetic field $\mathbf{B}_{act}$ (Fig. 1e and Supplementary Movie 4); for instance, as $B_{act}$ changed from 1.2 mT to 136 mT, the deformation angle $\alpha$ increased from 0° to 68° (Fig. 1f); meanwhile, mechanical deformation would induce dynamic change in inductance $\Delta L$ within the 2D MME coil skeleton of the butterfly robot (Fig. 1g and Supplementary Fig. 10), which can be measured by another high-frequency magnetic field (100 kHz) simultaneously.

Exploiting the magnetic actuation $\mathbf{B}_{act}$-mechanical deformation $\alpha$-inductance $\Delta L$ triangle as illustrated in Fig. 1i, we are able to construct a somatosensory 2D MME structure (designed with an integrated 2D MME coil) that can be actuated with an external magnetic field and senses deformation by measuring the inductance change due to the shape change of MME coil. Meanwhile, a variable external actuation magnetic field $\mathbf{B}_{act}$ also induces a dynamic change in the magnetic flux of the 2D MME coil, generating an induced voltage $E$ at the same output for inductance sensing, which can be utilized for wireless energy transmission. As shown in Fig. 1h and Supplementary Fig. 11, the output voltage $E$ increases approximately linearly with the strength of actuation magnetic field $\mathbf{B}_{act}$ at an actuation frequency of 2 Hz; when the actuation frequency increased to 110 kHz, the output induced voltage $E$ was enough to power LED lights (Supplementary Fig. 12 and Movie 4).

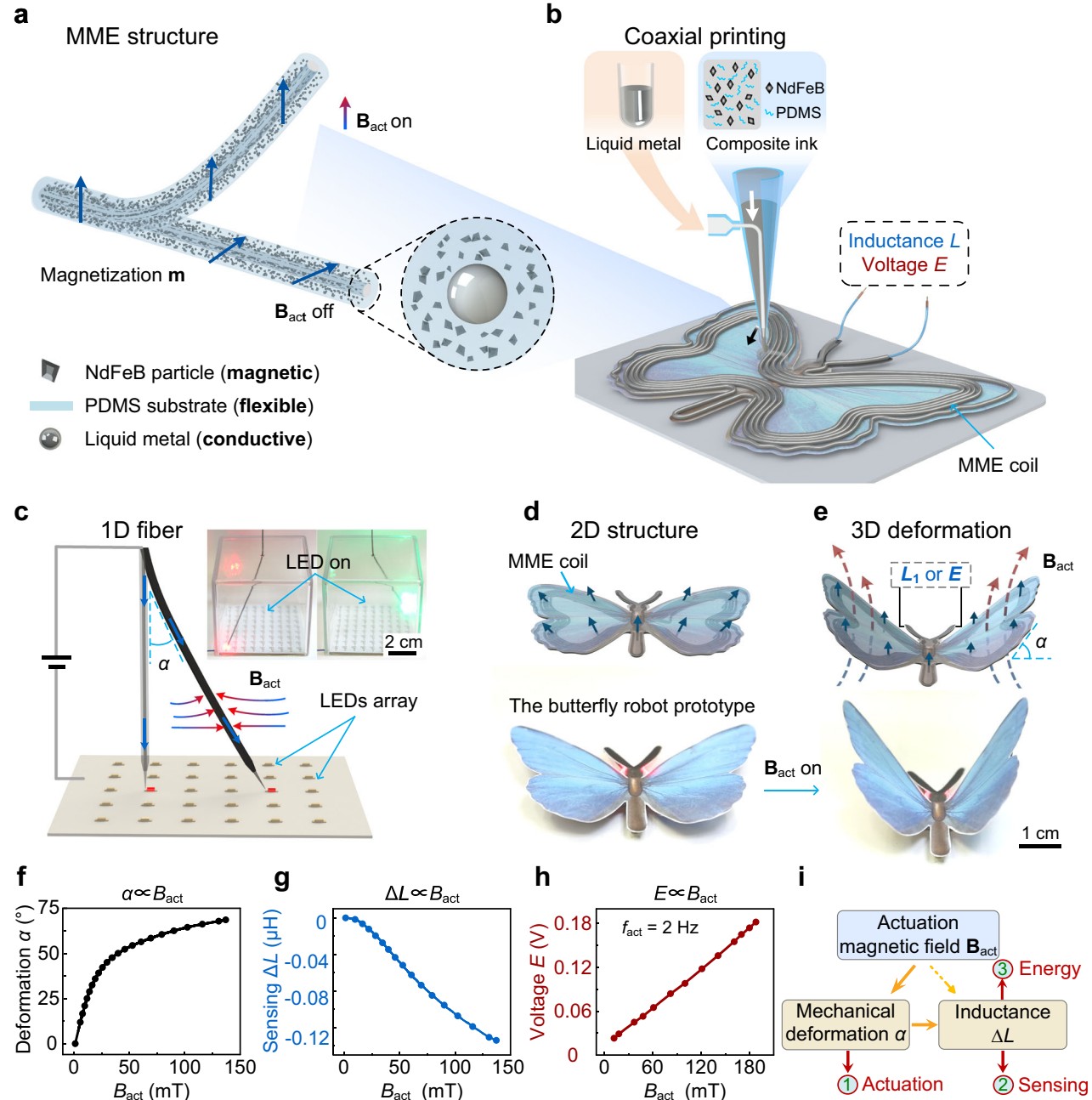

**Fig. 1 | Concept of the hybrid magnetic-mechanical-electrical (MME) structure.**
**a** Design and functional composition of the MME structure. $\mathbf{B}_{act}$ is the external
actuation magnetic field; $\mathbf{m}$ is the residual magnetization strength within the MME
structure. **b** Schematic of coaxial printing of the MME structure with coil geometry.
The MME coil structure acts as the skeleton of the butterfly robot. **c** Magnetic
actuation of the 1D MME fiber. Precise deformation of the 1D MME fiber can be
achieved with magnetic control for lighting LED pixels in a confined space.
**d** Schematic design of a 2D butterfly robot with the MME coil skeleton and the
fabricated prototype. **e** Illustration of the 3D deformation of the butterfly robot

actuated by the external actuation magnetic field $\mathbf{B}_{act}$. $\alpha$ is the deformation angle; $L_1$
is the inductance of the integrated MME coil in the deformed state; $E$ is the induced
voltage from the dynamic magnetic field. **f–h** Experiment results show the
dependence of mechanical deformation $\alpha$, inductance $\Delta L$, and the output voltage $E$
on the external actuation magnetic field $\mathbf{B}_{act}$. **i** The magnetic actuation $\mathbf{B}_{act}$-
mechanical deformation $\alpha$-inductance/voltage loop enabled by the MME coil
structure; the electrical inductive can be exploited either for deformation sensing
or wireless energy transmission.

## Fabrication of MME structures

In order to guide the coaxial printing of an MME fiber with a con-
tinuous core-sheath structure, two nondimensional parameters
($V^*$ and $K$) have been proposed. Figure 2a schematically illustrates the
key coaxial printing parameters, including the feed rate of liquid metal
$f_{liq}$, the feeding rate of PDMS @ NdFeB composite ink $f_{com}$, and the
printing speed $V$. We define a dimensionless feed rate $V^*$ ($V^* = f_{liq}/f_{com}$,
the $f_{liq}$ and $f_{com}$ are the feed rate of the liquid metal and the PDMS @
NdFeB composite ink, respectively) to describe the volume ratio of the

liquid metal core to the composite sheath per length. With reference to
$V^*$, the coaxial printing exhibits three printing behaviors (Fig. 2b, under
a constant printing speed $V = 18$ mm/s), including the discontinuous
state ($V^* < 0.6$), the thinning-normal state ($0.6 < V^* < 1.0$) and the
bursting state ($V^* > 1.0$). For $V^* < 0.6$, liquid metal was observed to
discontinuously disperse within the composite sheath, failing to gen-
erate a uniformly connected core due to insufficient feeding of the
liquid metal core (see the bottom left image in Fig. 2b). Appropriate feeding
rate of liquid metal core and composite sheath (i.e., $0.6 < V^* < 1.0$) can

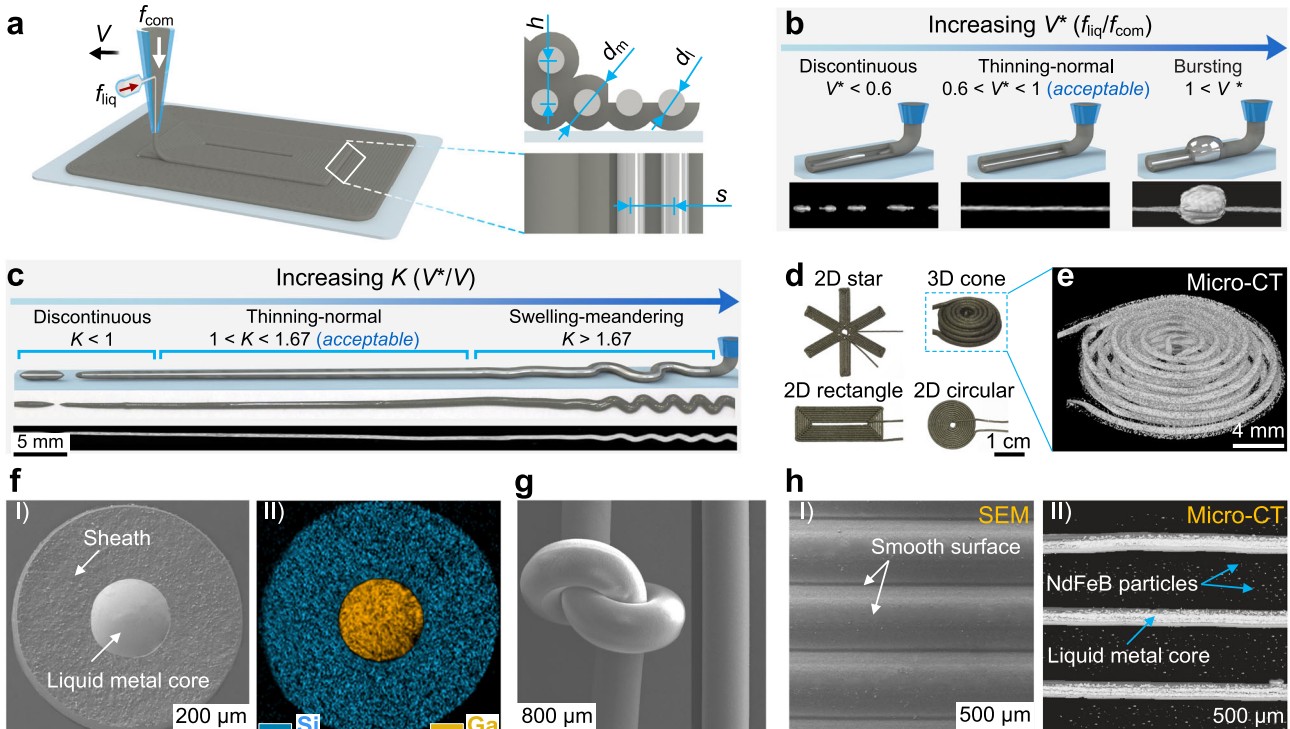

**Fig. 2 | Fabrication of the magnetic-mechanical-electrical (MME) structures.**
**a** Key parameters for coaxial printing. $f_{com}$: feed rate of the PDMS @ NdFeB composite ink to coaxial nozzle's outer channel; $f_{liq}$: feed rate of the liquid metal to the inner channel; $V$: printing speed; $d_m$ and $d_l$ are the outer diameter of the printed MME fiber and the diameter for the inner liquid metal core, respectively; $s$ and $h$ are the inter-fiber spacing and the inter-layer spacing, respectively. **b** Three typical printing states under different dimensionless feed rates $V^*$ ($V^* = f_{liq}/f_{com}$); corresponding micro-CT images of the MME fibers printed at different states are also provided; the liquid metal core can be clearly visualized in the micro-CT images **c** Morphologies of an MME fiber printed under different $K$ ($K = V^*/V$) at a constant $V^*$ of 1.0. **d** 2D/3D MME structures with complex geometries. **e** Micro-CT image of a multilayer 3D MME cone structure. **f** A SEM image shows the integral core-sheath structure of a typical MME fiber; the corresponding elemental mapping image is also provided. **g** SEM images demonstrating the flexibility of the MME fiber. **h** SEM images showing the smoothly connected MME fibers.

result in printing of a smooth core-sheath fiber with a continuous liquid metal core (see the bottom middle image in Fig. 2b). For $V^* > 1.0$, excessive liquid metal will rupture the composite ink sheath, forming liquid metal droplets along the printing path (see the bottom right image in Fig. 2b).

For high quality coaxial printing of the MME fiber with a continuous core-sheath structure, the printing speed $V$ needs to the compatible with the feeding rate of the composite ink and the liquid metal, which can be represented by a nondimensional parameter $K$ ($K = V^*/V$). For $K < 1$ (i.e., higher printing speed), the printed MME fiber will be over stretched, leading to a discontinuous fiber, as shown in Fig. 2c-left. A low printing speed (i.e., $K > 1.67$) would lead to insufficient stretching of the extruded uncured fiber, resulting in a serpentine structure with material accumulation (Fig. 2c-right). Thus, appropriate printing speed relative to the feeding rate (i.e., $1 < K < 1.67$) is required to form a continuous core-sheath MME fiber (Fig. 2c-middle). Figure 2c also shows that the diameter of printed core-sheath fiber can be tailored by the printing speed and the diameter of the coaxial nozzle. A fast printing speed and a smaller coaxial nozzle would result in smaller MME fiber in diameter.

Assisted by numerically controlled coaxial printing, the aforementioned 1D MME fiber can be easily used to fabricate the planner (2D) or the 3D MME structures, such as a planner star, a rectangle/circle, complex letters or a 3D cone (Fig. 2d–e and Supplementary Fig. 13). As the 2D MME structures are constructed by printing the MME fibers side by side, the inter-fiber spacing $s$ needs to be slightly smaller than the outer diameter of the MME fiber $d_m$ to ensure intimate contact between adjacently printed MME uncured fibers. A minimal inter-fiber spacing will cause inter-fiber squeezing, which may compress the core

and affect MME structures' conductivity. In contrast, larger inter-fiber spacing forms would generate isolated fibers without contact (Supplementary Fig. 14). Similarly, for layered 3D MME structures, the inter-layer spacing $h$ also needs to be slightly smaller than $d_m$ to ensure structural integrity and dimensional accuracy (Supplementary Fig. 15).

Under optimized printing conditions ($f_{liq} = 15$ mm/s, $f_{com} = 18$ mm/s, and $V = 18$ mm/s), the MME fiber after thermal curing exhibits a core-sheath structure (Fig. 2f, outer diameter $d_m$: about 830 μm, inner diameter for the liquid metal core $d_l$: about 270 μm) with excellent flexibility (Fig. 2g). In order to match the overall diameter of the printed fiber $d_m$ and control the width of the bonding area to 30 μm, the inter-fiber spacing $s$ was set as 800 μm for the construction of a 2D structure with a smooth external surface. (Fig. 2h and Supplementary Fig. 16).

The SEM image, EDS-based element mapping, and the micro-CT image all show a clear boundary between the NdFeB-PDMS composite sheath and the liquid metal core (Fig. 2f–g and Supplementary Fig. 17–19). Also, after printing (before curing), when the shear stress from printing is released, the composite sheath would exhibit solid-like behavior, which can effectively prevent the settling of the core liquid metal and help maintain the structural fidelity of the printed core-sheath structure. (Supplementary Fig. 20). Printing parameters for the 1D MME fiber in Fig. 2f–h were then used to fabricate samples for the investigation of mechanical, electrical, and magnetic properties of MME structures.

## Characterization of MME structures
MME fibers can be actuated by the external magnetic field due to the magnetization of NdFeB particles distributed in the sheath. Figure 3a shows the excellent ferromagnetic properties of a magnetic sheath

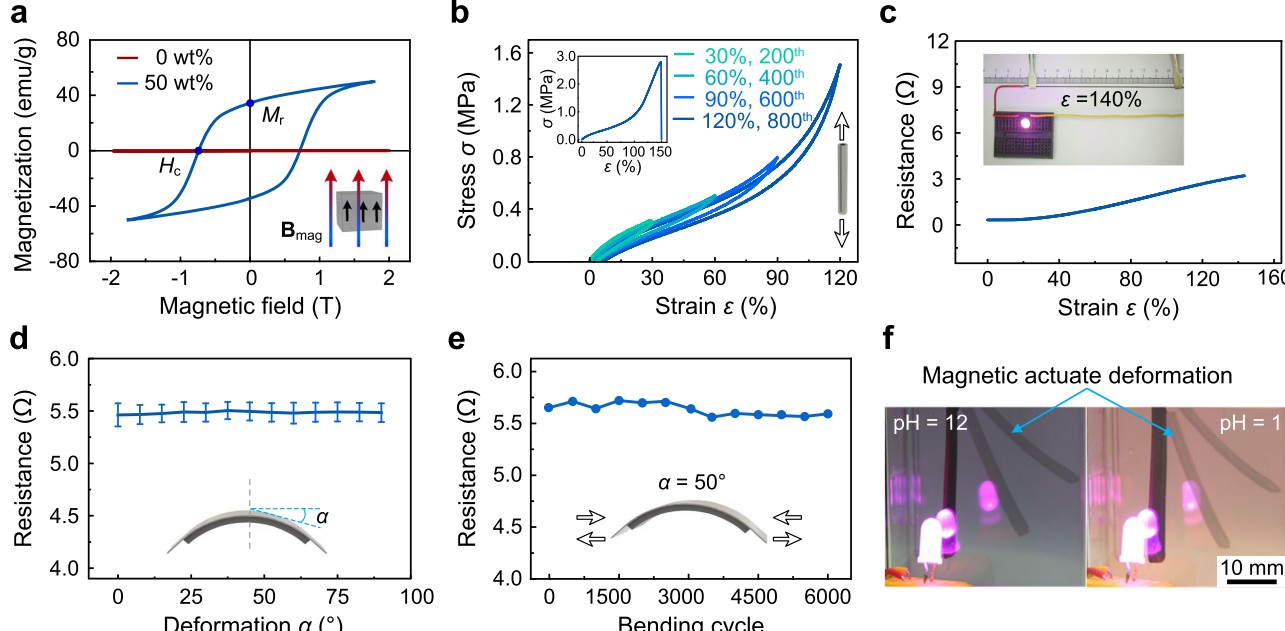

**Fig. 3 | Characterization of the magnetic-mechanical-electrical (MME) structures. a** Magnetic hysteresis loop for MME fiber's composite sheath with 50 wt% NdFeB (control: a pure PDMS sheath with 0 wt% NdFeB). **b** Cyclic loading-unloading test for the MME fiber; stretching rate: 0.5 mm/s; inter-cycle rest time: 10 min. **c** Effects of large deformation on electrical resistance; even at a large stretch of 140%, the MME fiber is still conducive to light a LED. **d** Effects of deformation angle $\alpha$ of the 2D MME structure (rectangle, length: 35 mm, width: 15 mm) on electrical resistance. Error bars are one standard deviation, and the number of independent experiments $n = 3$. **e** Durability of the 2D MME structure in response to different bending cycles. **f** Durability of the 2D MME structure immersed in acid (HCl) and alkaline (NaOH) solutions.

with the residual magnetization $M_r$, coercivity $H_c$, and relative permeability of 34 emu/g, 710 mT, and 1.02, respectively. Thus, the MME fiber/structure can be magnetized to saturation with a pulsed magnetic field (about 3 T), which is much larger than the saturation magnetization of NdFeB (about 1.7 T). The strength of the external actuation magnetic field $B_{act}$ (20-200 mT) is lower than the coercivity $H_c$ of NdFeB, which will not cause the re-magnetization.

Mechanical properties of the MME structure were characterized by tensile and cyclic loading-unloading tests using a motorized material testing system. As shown in Fig. 3b, the MME fiber exhibits excellent stretchability (stretched up to 150% strain). Reinforced by NdFeB micro-particles, the composite material is found to have a tensile strength of 2.88 MPa, much larger than that of the pure PDMS counterpart (tensile strength: 1.77 MPa, Supplementary Fig. 21). Also, loading-unloading tests for strain from 30 to 120% show that the MME fiber has excellent hyperplastic properties. At a large strain level of 120%, the maximum stress decreased by 12% after 1000 cycles' stretching (Supplementary Fig. 22e and f), which would be owing to the viscoelastic properties of the PDMS substrate[6]. But, at a moderate deformation (60% strain), the MME fiber showed stable mechanical behaviors (the maximum stress only reduced by 1.8%; Supplementary Fig. 22c and d). Also, after 1000 cycles' fatigue testing, the surface morphology of the MME fiber remained unchanged (Supplementary Fig. 22g), no micro-crack could be detected on the outer surface or the cross-section of the MME fiber, demonstrating the great durability of the MME fiber.

Electrical performance of the MME structure in response to deformation was also investigated. The electrical resistance of MME fiber (length: 165 mm, initial conductivity: about $2.07 \times 10^6$ S/m) increased from 0.35 Ω at 0% strain to 3.2 Ω at 140% strain due to the thinning of the liquid metal core during stretching (Fig. 3c and Supplementary Movie 5). Moreover, the MME fiber can maintain a constant electrical resistance in the stretched state (viz. maintained at 30, 60, 90, and 120 % strain for 300 s; resistance change <1%); in the cyclic fatigue test (120% strain), the electrical resistance was also stable (Supplementary Fig. 22a–b). This excellent electromechanical performance endows MME fiber with small resistance as electrical wires to power the LED even at 140% strain (inset in Fig. 3c). Moreover, the MME structures exhibit stable and robust electrical performance in response to bending to different angles (Fig. 3d) and cyclic bending over 6000 bending cycles (resistance variation <10% as shown in Fig. 3e and Supplementary Fig. 23).

Lastly, owing to the inert silicone matrix of the composite sheath, the MME structure can function in harsh environments. As shown in Fig. 3f and Supplementary Movie 6, after immersion in acid (pH = 1, HCl) or alkaline (pH = 12, NaOH) solutions for 3 h, the MME structure can still function properly in response to magnetic stimulation. With the demonstrated magnetic actuation, mechanical flexibility, electrical conductivity, and functional durability presented here, the MME structure can be used in a variety of applications ranging from soft robots to smart fabrics and wearable electronics[7,35].

## Magnetization for hybrid magnetic actuation, deformation and sensing

In order to achieve controllable deformation in response to an external actuation magnetic field $B_{act}$, a spatial magnetization profile **m** needs to be designed and imposed on the fabricated MME structure (Fig. 4a). To facilitate magnetization, a customized mold is desired to elastically deform the MME structure to a pre-designed 3D geometry; the MME structure fixed by the mold is then placed in a pulsed magnetic field $B_{mag}$ (Supplementary Fig. 5), allowing the magnetization of the NdFeB particles in the composite sheath along the direction of $B_{mag}$ field. After magnetization and release from the mold, the MME structure has a spatial magnetization profile **m** $(x, y, z)$ across its surface. The magnitude of **m** $(x, y, z)$ is mainly influenced by the content of magnetic particles in the composite material while its profile is controlled by the mold geometry and the relative orientation between the local curvature of the mold and the direction of $B_{mag}$ field.

To facilitate rational design of the magnetization profile **m**, a finite element analysis (FEA) model has been established (Supplementary

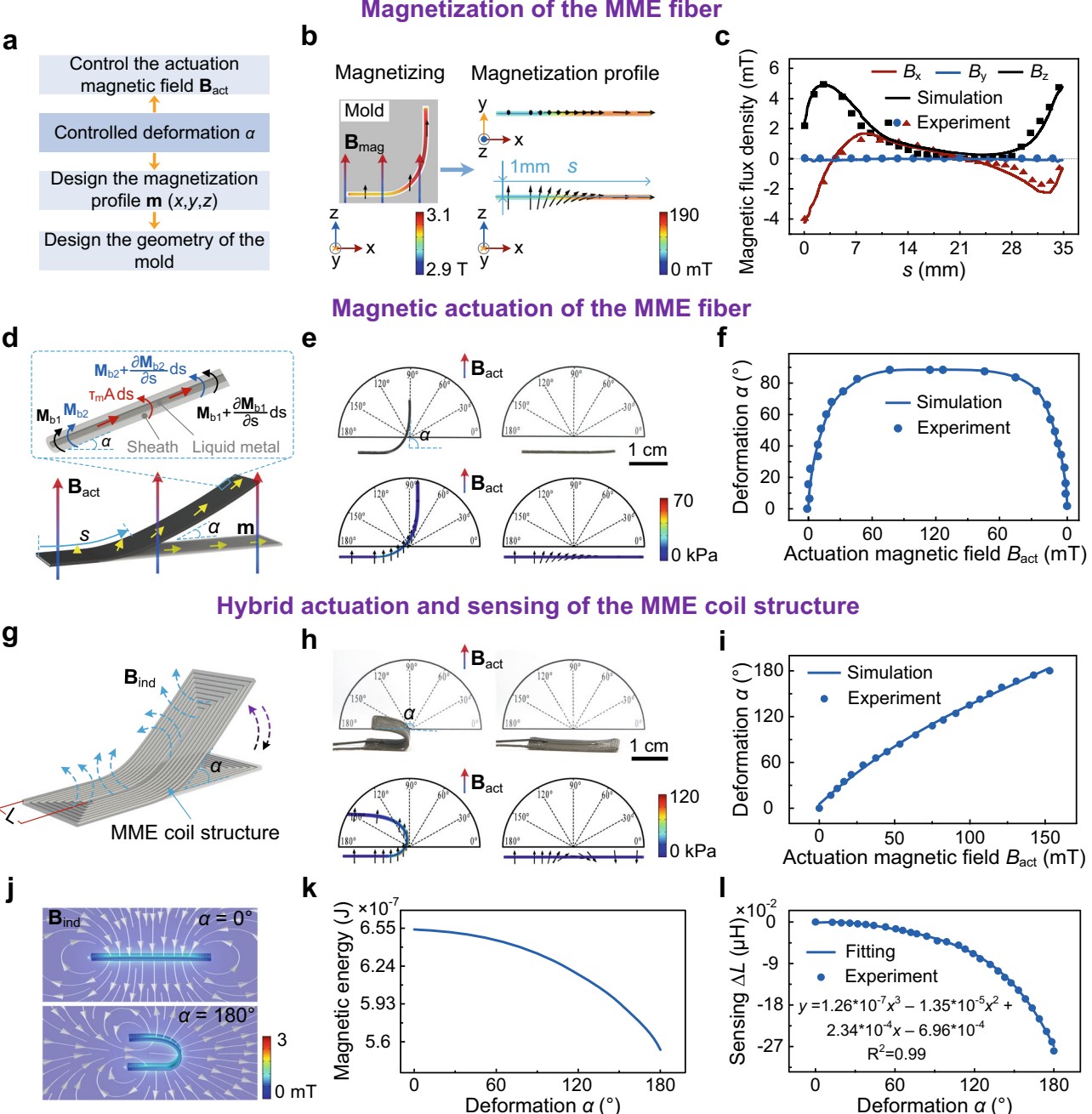

**Fig. 4 | Magnetization, magnetic actuation, and sensing of the magnetic-mechanical-electrical (MME) structures. a** Strategies for realizing controllable magnetically actuated deformation of MME structures. **b** Simulation predictions of the magnetization process of the MME fiber. The black arrows represent the magnetization profiles. **c** Experimental and simulation results of the residual magnetization profiles **B** ($B_x$, $B_y$, $B_z$) along the central axis $s$ of the MME fiber. **d** Quasi-static analysis of the MME fiber. The bending moment acting on an infinitesimal element of the MME fiber under steady-state deformation is shown in the inset. **e** Simulation prediction and experimental results of magnetically actuated deformation of the MME fiber. **f** Deformation angle $\alpha$ of the MME fiber in response to the actuating magnetic field **B**$_{act}$. **g** Schematic of hybrid actuation and sensing functions of the MME coil structure (length: 35 mm, width: 15 mm). **h**, **i** Simulation prediction and experimental results of magnetically actuated deformation of the MME coil structure. **j** Simulation analysis of the inductance magnetic field **B**$_{ind}$ of coil change caused by the deformation of the MME coil structure ($\alpha = 0°$ and $\alpha = 180°$). **k** The variation of magnetic energy of the MME coil structure with deformation angle $\alpha$ calculated by simulation. **l** The variation value of inductance $\Delta L$ predicted from simulation and experimental measurements plotted in response to the deformation angle $\alpha$.

Fig. 24–25 and Movie 7). The magnetization profile **m** corresponding to the pre-designed 3D geometry of the MME structure can be calculated. In this way, the mold structure can be optimized through FEA. To validate the predictions from simulation analysis, the magnetization profile was measured by a gaussmeter at 1 mm off the surface of the MME structure and represented as magnetic flux density **B** ($B_x$, $B_y$, $B_z$) (Supplementary Fig. 26). As shown in Fig. 4b and c, the theoretically

predicted magnetization profiles in *x-y*, *x-z* and *y-z* planes of the MME fiber agree well with the experimental measurements (magnetization of MME fiber assisted by a L-shaped mold, Supplementary Fig. 27).

Using the coaxial printing technique and the customized magnetization method, we are able to design and fabricate 2D/3D MME structures $S$ ($x$, $y$, $z$) imposed with 2D/3D magnetization profiles **m** ($x$, $y$, $z$). For an MME structure with a pre-defined mechanical structure

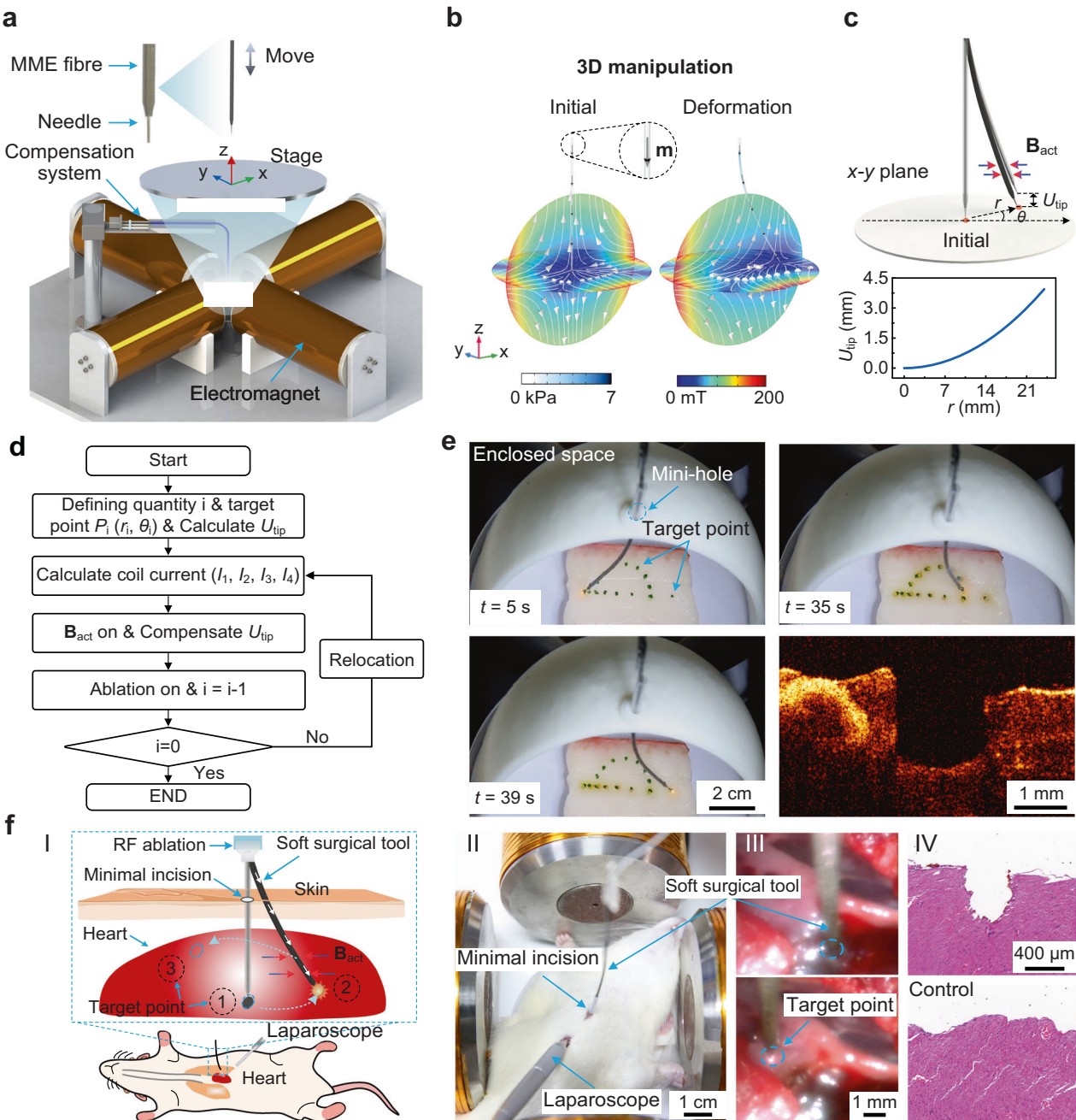

**Fig. 5 | Minimally invasive electro-ablation surgery capabilities of a 1D magnetic-mechanical-electrical (MME) fiber. a** Schematic of a catheter-style soft surgical tool that can be controlled by a four-electromagnets magnetic navigation system. **b** Actuation of the catheter-style soft surgical tool; the magnetic field generated by the magnetic navigation system in the initial and deformed states of the catheter-style soft surgical tool. The black arrows represent the magnetization profile of the surgical tool. The white arrows represent the actuating magnetic field vectors. **c** Tip deviation in the vertical direction $U_{tip}$ as the tip of the catheter-style soft surgical tool moves from the initial position to a target point $P(r, \theta, z)$ on the $xy$ plan ($z = 0$). The graph shows $U_{tip}$ as a function of radius $r$. **d** Control procedure of the electro-ablation surgery under magnetic control. **e** The catheter-style soft surgical tool performs precise ablation of predefined points (the minimum distance is 1 mm) on porcine tissue in an enclosed space under the control of an actuation magnetic field. **f** Illustration of in vivo electro-ablation surgery with the catheter-style soft surgical tool (I); (II) the experimental setup for the in vivo experiment; (III) the process of minimally invasive ablation on the heart surface of male Sprague-Dawley (SD) rats; (IV) H&E staining of heart tissue after electro-ablation surgery. Control: without electro-ablation surgery.

$S(x, y, z)$ and a customized magnetization profile $\mathbf{m}(x, y, z)$, controllable deformation can be achieved by adjusting the actuation magnetic field $\mathbf{B}_{act}(x, y, z)$. Interaction between $\mathbf{m}(x, y, z)$ and $\mathbf{B}_{act}(x, y, z)$ will generate a spatially varying magnetic torque across the MME structure ($\mathbf{T}_m = \mathbf{M}_{net} \times \mathbf{B}_{act}$, $\mathbf{M}_{net} = \int_0^S \mathbf{m}A ds$). As explained in Fig. 4d, in response to $\mathbf{B}_{act}$ field, the MME structure will be deformed to an equilibrium state, resulting from the balance between the magnetic torque $\mathbf{T}_m$ and torque $\mathbf{T}_e$ from the accompanying elastic deformation

(Supplementary Fig. 28). To predict the magnetically actuated deformation, a finite element model was also developed. Figure 4e, h and Supplementary Fig. 29 compare theoretically predicted and experimentally observed deformations of a 1D MME fiber and a 2D MME planar structure (with or without magnetic actuation). As validated in Fig. 4f and i, the theoretical model can accurately capture the deformation behavior of MME structures, laying a solid foundation for accurate and controllable magnetic actuation.

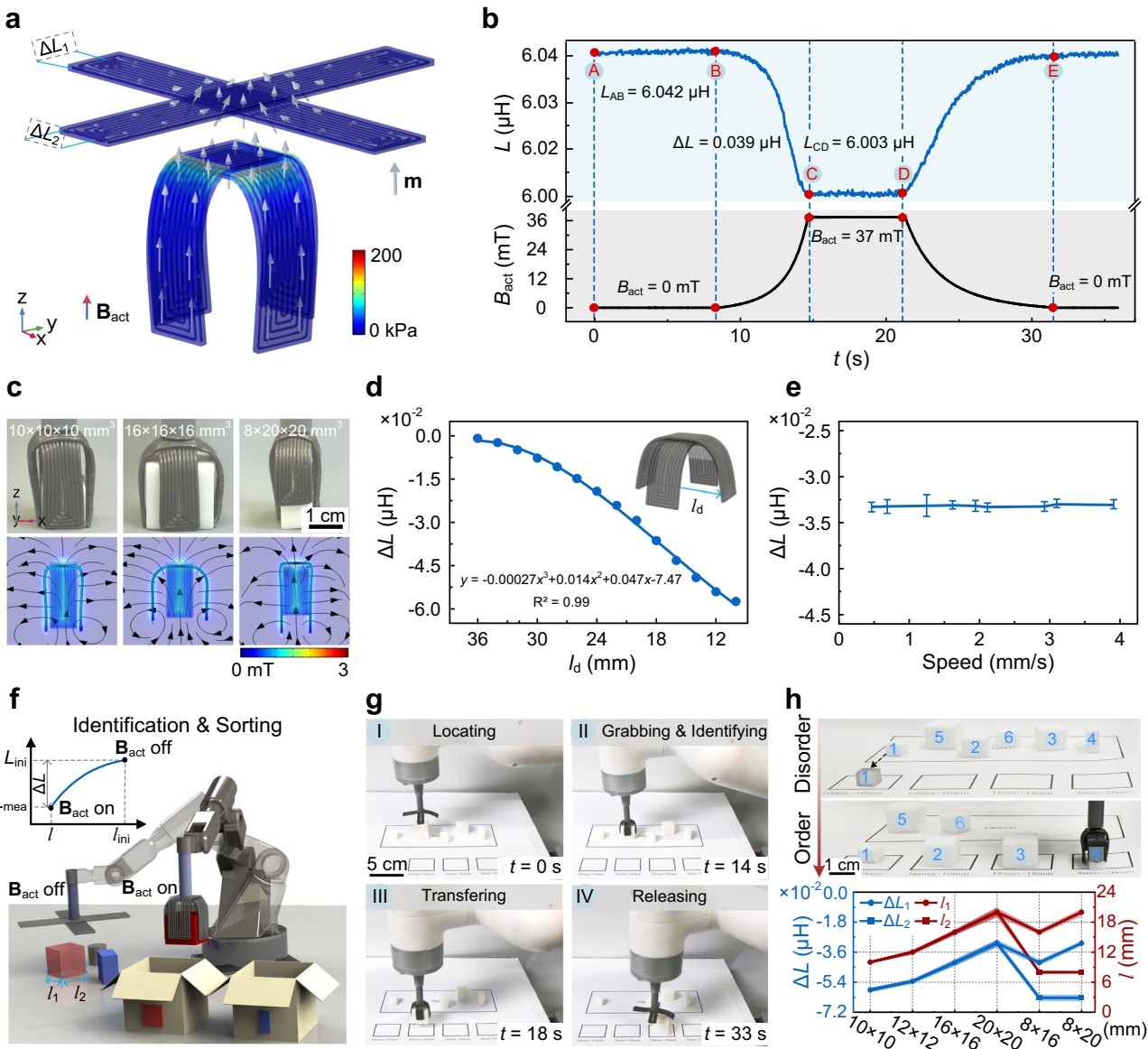

**Fig. 6 | Hybrid deformation and sensing capabilities of a 2D magnetic-mechanical-electrical (MME) gripper. a** Simulation results of the deformation of an MME gripper with integrated coils actuated by an external magnetic field. The white arrows represent the magnetization profile **m** of the MME gripper. **b** Inductance $L$ time curve for the MME gripper and the corresponding actuation magnetic field strength $B_{act}$ time curve during object grasping, transferring, and releasing. **c** Deformation of the MME gripper grasping objects of different sizes (top) and the corresponding induced magnetic field $B_{ind}$ (bottom). The $B_{ind}$ field is generated by an AC voltage (2 V, 100 kHz) applied on the MME coil during grasping. **d** The variation of inductance $\Delta L$ of an MME gripper with the distance between the two edges of the gripper $l_d$. **e** The variation of inductance $\Delta L$ of an MME gripper in response to different deformation speeds at $l_d = 19$ mm. Error bars are one standard deviation, and the number of independent experiments $n = 3$. **f** Schematic of the robot-assisted MME gripper for object identification/sorting. **g** The process of grasping and recognizing objects by the robot-assisted MME gripper. **h** The MME gripper sorts cluttered objects (top). The variation of inductance $\Delta L$ and the corresponding perceived object size $l$ of the gripper during the sorting process (bottom). Error bars are one standard deviation, and the number of independent experiments $n = 3$.

If a 2D MME structure is further designed with an MME coil, deformation of the MME coil structure under magnetic actuation will also deform the coil and change the inductance of the MME coil, which can be used for in-situ deformation sensing by measuring the inductance change. For inductance measurements, a high-frequency AC voltage signal (2.0 V, 100 kHz) can be applied on the MME coil as illustrated in Fig. 4g, generating an inductive magnetic field $B_{ind}$ in the MME coil. As shown in Fig. 4j and Supplementary Movie 7, the inductance magnetic field $B_{ind}$ is affected by the coil deformation. At a large deformation angle $\alpha$, the $B_{ind}$ field generated by the two halves of the deformed coil will increasingly repel and cancel each other, leading to a decreased magnetic energy stored in the deformed coil (Fig. 4k). As

the magnetic energy is proportional to the inductance[65], inductance is found to decrease with the increase of the deformation angle as experimentally observed in Fig. 4l. The quantitative relationships among actuation magnetic field $B_{act}$, mechanical deformation $\alpha$ and the inductance $\Delta L$ ($B_{atc}$-$\alpha$ in Fig. 4f and i, $\Delta L$-$\alpha$ in Fig. 4l) therefore enables somatosensory actuation in the MME coil structure.

Furthermore, when the 2D MME coil structure is placed in a closed circuit, the changing external actuation magnetic field $B_{atc}$ causes the magnetic flux $\Phi$ of the 2D MME coil structure to change, generating an induced voltage $E$ in the coil. This is in accordance with the Faraday law of electromagnetic induction: $E(t) = -n(d\Phi/dt)$. The induced voltage increases with increasing magnetic field strength and frequency.

Therefore, the 2D MME coil structure can convert the changing magnetic field into electrical energy for wireless power transmission, which will be discussed later in the wireless energy transmission capability of the soft MME robot.

## A flexible MME fiber for minimally invasive electro-ablation surgery

Controllable deformation of a 1D MME fiber enables application in minimally invasive surgery with magnetic actuation. Figure 5 demonstrates a miniaturized catheter-style soft surgical tool composed of an MME fiber magnetized along its axis (diameter: 0.83 mm, length: 75 mm) and a microneedle (diameter: 0.35 mm, length: 5 mm), which can be used for electro-ablation surgery under magnetic control in a confined space. For precise control, a navigation system (Supplementary Fig. 30) with four electromagnets has been constructed to generate a controllable magnetic field in the operating space by adjusting the current (radius $r$: 0–22.5 mm, angle $\theta$: 0–360°; Fig. 5a, Supplementary Fig. 31 and Movie 8). The origin of the global cylindrical coordinate system $P_n$ ($r_n$, $\theta_n$, $z$) is placed at the center of the four symmetrical electromagnets. By adjusting the electrical current in the four electromagnets, a magnetic field can be generated in the operating space with the zero magnetic strength point ($B_{act} = 0$ mT) as the origin of the magnetic field.

In the initial state (Fig. 5b-left), the catheter-style soft surgical tool is vertically aligned with the $z$-axis; the tip of the tool and the zero magnetic strength point are both located at the origin of the global coordinate system. By adjusting the current in the four electromagnets, the position of the zero magnetic strength point can be controlled in the $x$-$y$ plane (Fig. 5b-right). As previously explained, due to the equilibrium between the magnetic force and the body force from elastic deformation, the MME fiber would deform in the magnetic field generated in the operating space, reaching an equilibrium state with the tip of the catheter-style soft surgical tool located at the zero magnetic strength point. Therefore, by adjusting the position of the zero magnetic strength point in the $x$-$y$ plane, the tip of the surgical tool can be maneuvered in the operating space. In the magnetically actuated deformation process, an auxiliary numerically controlled stage is used to assign a compensating up-down motion (along the $z$-axis) to the catheter-style soft surgical tool, keeping the tip of the catheter in the $x$-$y$ plane; otherwise, the deformation would lead to a vertical tip deviation of $U_{tip}$ as illustrated in Fig. 5c.

The effectiveness of the catheter-style soft surgical tool for minimally invasive electro-ablation surgery (Fig. 5d) has been verified in vitro on porcine tissue in a confined space mimicking the abdominal cavity (Fig. 5e and Supplementary Movie 8). The surgical path was pre-operatively planned according to the predefined target points on the tissue surface. The catheter-style soft surgical tool is inserted into the enclosed space through a keyhole (about 2.5 mm in diameter). For each target point, the electrical current for the four electromagnets and the corresponding compensation motion by the numerically controlled stage was calculated in advance. Following the pre-defined path, electrical ablation could be performed on the tissue surface in the dot-mode (Fig. 5e) or the continuous mode (Supplementary Fig. 32a). Tissue ablation made on the tissue surface could be clearly visualized by OCT (bottom-right in Fig. 5e; Supplementary Fig. 32b).

In vivo minimally invasive electro-ablation surgery with magnetic navigation is further performed on rat skin and heart (Fig. 5f-I, Supplementary Fig. 33 and Movie 8). In the skin ablation test, an anesthetized male Sprague-Dawley (SD) rat is placed in an enclosed space on the operating space for electro-ablation on the skin (Supplementary Fig. 33a). After skin ablation, skin tissue is harvested and stained with hematoxylin and eosin (H&E) assays; as shown in the histological images in Supplementary Fig. 33b, controllable tissue ablation was successfully achieved. For the heart surgery, the catheter-style soft surgical tool is introduced into the rat's chest cavity through a keyhole;

a digital laparoscope (diameter 5 mm) inserted through another keyhole is used to record the ablation process (Fig. 5f-II). Details of the ablation experiments are found in the experimental section. As shown in Fig. 5f-III and Supplementary Movie 8, the catheter-style soft surgical tool could be navigated to access different targets on the heart surface through a single entrance hole on the abdominal wall for controlled minimally invasive electro-ablation surgery (Fig. 5f II–IV). Compared with the operation by a rigid interventional device, the catheter-style soft surgical tool offers the flexibility of accessing a large operation area in a confined space through a small entrance hole, presenting a promising strategy for targeted and minimally invasive surgery.

## A magnetically actuated MME gripper with in-situ somatosensory capability

2D MME structures with integrated MME coils further enable synchronous actuation and deformation sensing in a single structure, which is not commonly reported in the literature[66–70]. Exploiting the hybrid somatosensory actuation properties, a magnetically actuated MME gripper has been developed. As shown in Fig. 6a, the MME gripper is designed with two orthogonally overlapped rectangular MME coils (length: 51 mm, width: 9 mm; Supplementary Fig. 34). With the magnetization method described in Fig. 6a, the MME gripper was customized with a magnetization profile **m** shown in Fig. 6a-top (see the magnetization process in Supplementary Fig. 35). After magnetization, this MME gripper can be magnetically deformed and actuated by the actuation magnetic field **B**$_{act}$ to grasp objects of different sizes (Fig. 6c and Supplementary Fig. 36). During grasping or releasing, the dynamically changing inductive magnetic field **B**$_{ind}$ of the deformed MME gripper (generated by another external sensing AC voltage signal applied to the MME coils; Fig. 6c and Supplementary Movie 9) would alter the inductance for the two MME coils ($\Delta L_1$ and $\Delta L_2$). The quantitative relationship between $\Delta L$ and $l_d$ ($l_d$ is the distance between the two edges of the MME coil) as shown in Fig. 6d, thus enables the MME gripper to feel the size of the grasped object. For instance, as $l_d$ decreased from 36 mm to 10 mm, the $L$ decreased from 6.037 µH to 5.981 µH (Supplementary Fig. 37a).

With the hybrid actuation and sensing functions, the MME gripper can feel the grasping contact and release of an object. Figure 6b shows the dynamic change of the inductance $L$ during object grasping, transferring, and releasing. At point B, the inductance $L$ starts to decrease, implying that the MME gripper is being deformed for grasping. At point C, the inductance $L$ stabilizes to a constant value, indicating that the gripper has firmly grasped the object. At point D, the inductance $L$ abruptly increases, implying that the object is released by the gripper. At point E, the inductance $L$ returns to its initial value, showing the gripper has returned to its deformation-free state. Exploiting the inductive-time curve as shown in Fig. 6b, close-loop grasping and sensing can be achieved with a single actuator. Also, as shown in Fig. 6e and Supplementary Fig. 37b, the relative change in inductance $\Delta L$ for a constant $l_d$ is independent of the grasping speeds from 0.5 mm/s to 4.0 mm/s, revealing that the MME gripper can stably sense the object size independent of the grasping speed.

Integrated with a robotic arm, the MME gripper can be used for complex tasks such as object identification/sorting. A control procedure was also developed for this purpose (Fig. 6f and Supplementary Fig. 38). The process for robot-assisted object grasping is shown in Fig. 6g and Supplementary Movie 9. The robotic arm can locate the gripper for object grasping; the actuation magnetic field **B**$_{act}$ can be applied to actuate the gripper for grasping or removed for object releasing (see $B_{act}$-$t$ curve in Fig. 6b). Further, by measuring the inductance change in-situ ($\Delta L_1$ or $\Delta L_2$), the gripper can detect the object size ($l_1$ or $l_2$, Fig. 6f) at the moment of grasping, which can be utilized for object sorting (Fig. 6h). Thus, with the hybrid actuation and sensing capabilities, the MME gripper can classify and sort chaotically placed objects into different groups according to their dimensions

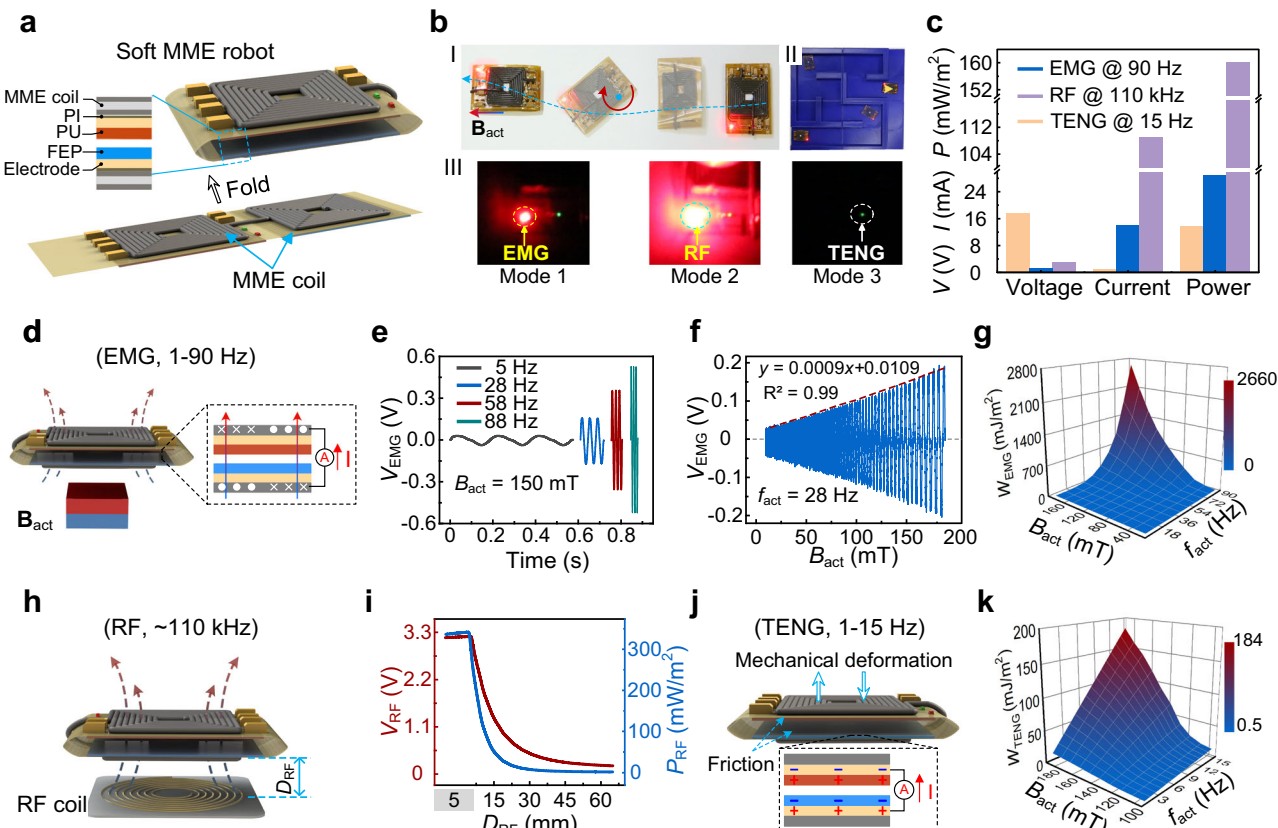

**Fig. 7 | Hybrid actuation and wireless energy transmission enabled by a soft magnetic-mechanical-electrical (MME) robot. a** Design of the soft MME robot. **b** Magnetically controlled rotation and translation of the soft MME robot. Actuated by a rotating magnetic field and a gradient magnetic field, the soft MME robot performs rotational and translational motions (I) and passes through the maze (II). (III) Along with the motion/deformation, the soft MME robot can generate energy in three modes (Mode 1: low-frequency electromagnetic power generation to light up the red LED; Mode 2: high-frequency electromagnetic power generation to light up the red LED; Mode 3: triboelectric power generation to light up the green LED). **c** Comparison of power generation capabilities in the three modes (voltage $V$,

current $A$, power $P$). **d** Wireless energy transmission with an electromagnetic generator (EMG). **e**–**g** Electrical output performance of the EMG; (**e**) output voltage under different working frequencies (5–88 Hz), (**f**) output voltage with different magnetic field strengths (5–190 mT), (**g**) output energy within 1 s at different frequencies and magnetic field strengths. **h** Wireless energy transmission by the radio frequency (RF). **i** Electrical output performance (output voltage and output power) of RF under different working distances ($D_{RF}$) (range: 5–65 mm). **j** Schematic illustration of the working principle of a triboelectric nanogenerator (TENG). **k** Output energy within 1 s with different frequencies and strengths of the magnetic field.

(Fig. 6h). The maximum gripping weight would be affected by both the object geometry/shape and the pressure applied onto the object by the gripper (which in turn is affected by the strength of the magnetic field; Supplementary Fig. 39a–b). In addition, after cyclic gripping tests, the MME gripper still had stable magnetic properties and lifting capabilities (Supplementary Fig. 39c–d). Compared with other manipulators/actuators[66–70] (Supplementary Table 3), the MME gripper developed in this study enables wireless magnetically actuated deformation and hybrid in-situ sensing with a simple structure. As demonstrated here, this hybrid actuation and sensing mechanism has the potential to bring new capabilities to soft robots.

**A soft MME robot with wireless energy transmission capability**
The coaxially printed MME coil structure can be further utilized for wireless energy transmission along with magnetically actuated deformation. To demonstrate this potential application, a soft robot was designed with two MME coils on a flexible substrate with integrated circuits as components of an electromagnetic generator (Fig. 7a). Besides the electromagnetic generator, a triboelectric nanogenerator (TENG) was also designed on soft MME robot's inner surfaces, with fluorinated ethylene propylene (FEP) and polyurethane (PU) as the triboelectric pair (Fig. 7a; see Supplementary Fig. 40 for the rectifier circuit). For the magnetized soft MME robot (see the magnetization profile **m** in Supplementary Fig. 41), rotation of the actuation magnetic

field $\mathbf{B}_{act}$ generates a torque, causing the soft MME robot to rotate. Likewise, a gradient in the actuation magnetic field $\mathbf{B}_{act}$ causes the soft MME robot to move along the gradient direction (Fig. 7b-I and Supplementary Movie 10). Thus, by controlling the external magnetic field, the soft MME robot can be maneuvered through a complex route (Fig. 7b-II and Supplementary Movie 10). In the magnetically actuated rotation/translation process, energy generation can be achieved via three different modes (Fig. 7b-III), e.g., wireless energy transmission by electromagnetic generation (EMG, Mode 1, low frequency; Fig. 7d), radio frequency wireless energy transmission (RF, Mode 2, high-frequency; Fig. 7h) and energy generation by TENG (Mode 3, low frequency; Fig. 7j).

As for Mode 1 (the EMG mode), the energy is generated based on the Faraday law of electromagnetic induction (Fig. 7d). Energy generation in Mode 1 can produce an open-circuit peak voltage $V_{EMG}$ of 0.16 V, a short-circuit peak current $I_{EMG}$ of 14.1 mA, and a peak power density $P_{EMG}$ of 28.9 mW/m² at an external resistance load of 390 Ω ($B_{act}$ = 165 mT, $f_{act}$ = 24 Hz; Supplementary Fig. 42). As power generation by EMG is mainly affected by the magnetic flux change rate, the output voltage $V_{EMG}$ is found to be affected by both the frequency (Fig. 7e) and the actuation magnetic field strength (Fig. 7f). For instance, as the frequency increases from 5 Hz to 88 Hz, the $V_{EMG}$ increases from 0.027 V to 0.52 V (Fig. 7e). As the magnetic field strength increased from 10 mT to 185 mT, the $V_{EMG}$ increased from

0.027 V to 0.19 V (Fig. 7f). Correspondingly, as shown in Fig. 7g, with the increase of magnetic field strength and frequency, the output capacity of EMG per unit time increases (maximum $W_{EMG}$ = 2660 mJ/m$^2$, $t$ = 1 s).

As for Mode 2 (the RF mode), the energy is also generated based on the principle of electromagnetic induction (Fig. 7h). As power generation by RF is mainly affected by the working distances $D_{RF}$, the output voltage $V_{RF}$ and $P_{RF}$ are found to be affected by the working distances $D_{RF}$ (Fig. 7i). For instance, at $D_{RF}$ = 5 mm, $V_{RF}$ and $P_{RF}$ remain unchanged at 3.1 V and 335 mW/m$^2$, respectively. As $D_{RF}$ gradually increases to 65 mm, $V_{RF}$ and $P_{RF}$ decrease to 0.19 V and 1.19 mW/m$^2$, respectively. The peak power density of RF is 163 mW/m$^2$ at an external resistance load of 19 $\Omega$ ($D_{RF}$ = 28 mm, Supplementary Fig. 43).

As for Mode 3 (the TENG mode), energy is generated due to the contact/separation of the triboelectric pair under magnetic actuation (Fig. 7j). Energy generation in Mode 3 can produce an open-circuit peak voltage $V_{TENG}$ of 18 V, a short-circuit peak current $I_{TENG}$ of 1.2 μA, and a peak power density $P_{TENG}$ of 13.8 mW/m$^2$ at an external resistance load of 42 MΩ ($B_{act}$ = 180 mT, $f_{act}$ = 1 Hz; Supplementary Fig. 44). Effects of actuation frequency $f_{act}$ and strength of the actuation magnetic field $B_{act}$ on the output voltage $V_{TENG}$ and power $W_{TENG}$ are summarized in Fig. 7k and Supplementary Fig. 45. The output voltage $V_{TENG}$ is found to be minimally affected by the actuation frequency (Supplementary Fig. 45a), while a stronger actuation magnetic field $B_{atc}$ could help generate a larger voltage $V_{TENG}$ (Supplementary Fig. 45b). As for the output energy, a larger $B_{atc}$ and a higher actuation frequency $f_{act}$ would collectively increase the output power in the TENG mode.

Figure 7c further compares the wireless energy transmission performance for the soft MME robot operating in these three modes. For instance, the high-voltage power generation (18 V) under the action of a low-frequency magnetic field (15 Hz) and high-power generation (160 mW/m$^2$) under the action of a high-frequency magnetic field (110 kHz). Wireless energy transmission enabled by an MME coil has the potential to solve the energy-limited problems of small-sized magnetically controlled soft robots.

## Discussion

Integration of hybrid functions including magnetically actuated deformation, electrical conduction, in-situ somatosensory, and energy harvesting in a single structure have been challenging for magnetoactive materials and devices. In this work, we achieve successful integration with a magnetic-mechanical-electrical (MME) core-sheath fiber structure composed of a liquid metal (EGaIn) core surrounded by a soft magnetoactive composite sheath (NdFeB @ PDMS composite). The unique core-sheath MME structure can intimately combine magnetic responsiveness, high electrical conductivity ($2.07 \times 10^6$ S/m), robust mechanical properties (150% strain limit and 0.87 MPa Young's modulus), and functional durability (remain highly conductive in tension and bending and functions in harsh environments) in a single structure. Assisted by numerically controlled coaxial printing with optimized printing parameters, we are able to fabricate electromagnetic devices with complex MME structures in one step. In particular, the MME coil structure allows for closed-loop coupling of magnetic actuation, mechanical deformation, electrical conductivity, and electromagnetic inductance, which can be exploited for hybrid magnetic actuated deformation, in-situ somatosensory and even wireless energy transmission.

To demonstrate the potential of this material design strategy, we first designed a catheter-style soft surgical tool (1D MME fiber) for minimally invasive electro-ablation surgery in vivo on a rat heart. Controlled by an actuation magnetic field, the flexible MME fiber catheter can be manipulated in an enclosed space for precise and minimally invasive surgery on rats. Furthermore, a soft somatosensory gripper with integrated MME coils was developed, which was

capable of hybrid actuation and sensing; during magnetically actuated gripping, the gripper can feel the touch/release of the object and sense the object size; integrated with a robotic arm, this soft MME somatosensory gripper can be controlled to perform complex tasks such as object identification or sorting. To demonstrate the hybrid actuation and energy harvesting functionality, a soft robot with 2D MME coil structures was developed, which could be maneuvered through a complex maze-like pathway with translational/rotational motions under the remote control of an actuation magnetic field. Moreover, owing to the MME coil structures integrated in the soft robot, wireless and multimodal energy transmission/harvesting can be achieved along with robot motion/deformation. Together, these examples demonstrate the ability to combine hybrid, multi-modal functionality (i.e., magnetically actuated deformation, electric conduction/sensing, and wireless energy transmission) in a single soft printed structure. Moreover, they show how this material architecture could enable new capabilities for biomedical devices or soft robots that require mechanical compliance, compact design, and versatile multifunctionality.

## Methods
### Preparation of the printing ink
The composite ink was prepared by adding NdFeB microparticles (average diameter: 5 μm; MQP-15-7, Magnequench, China) to PDMS (SE 1700, Dow Corning, USA) at a weight ratio of 1:1 (Supplementary Fig. 4). Then mixed thoroughly by mechanical mixer (RW20, IKA, Germany) at 2000 r.p.m. for 3 min, followed by degassing in a vacuum chamber for 10 min. Finally, a homogeneous curable composite ink was prepared for coaxial printing. Liquid metal (Ga, 75.5%, and In, 24.5% by weight; melting point, about 16 °C) was purchased from Dongguan Dingguan Metal Technology Co., LTD.

### Rheological characterization
Rheological responses of the composite ink and liquid metal were characterized using a stress-controlled rheometer (HAAKE Mars III, Thermo Scientific, USA). Apparent viscosities of composite ink and liquid metal were measured via steady-state flow experiments with a sweep of shear rates (0.003-1000 s$^{-1}$). Shear storage moduli of composite ink were measured as a function of shear stress via oscillation experiments at a fixed frequency of 1 Hz with a sweep of stress (4.6-191465 Pa·s). The composite ink was equilibrated at 25 °C for 10 min before testing, and all experiments were performed at 25 °C with a gap height of 0.5 mm.

### Coaxial printing procedure
A custom-designed coaxial printer with the coaxial extrusion mechanism was developed to print MME structures, as shown in Supplementary Fig. 3. Models of MME structures were first built using SolidWorks (Version 2016, Dassault Systemes, France) and converted into G-codes using repetier-host (Repetier, Germany). The composite ink and liquid metal in the two syringes were extruded through a customized nozzle (inner tube diameter $d_i$: 380 μm; nozzle diameter $d_o$: 800 μm) to prepare core-sheath fiber. Synchronously extrude and move the nozzle according to the designed code to print complex MME structures. The typical coaxial printing processes of MME structures are shown in Supplementary Movie 1. Coaxial printed MME structures were cured at 70 °C for 1 h in an oven. In order to demonstrate the stability of the printing performance, a 3-layer fiber structure was printed with two different sheath materials (pure PDMS: 0 wt% NdFeB in PDMS; the composite ink: 50 wt% NdFeB in PDMS; both inter-fiber spacing and inter-layer spacing are 830 μm). After printing, an object (cubic; size:10 mm by 10 mm) was placed on top of the printed uncured structure mimicking the weights exerted on bottom layers by the upper layers in the curing process (Supplementary Fig. 20). Weights of the object were adjusted to mimic 30 or 100 layers fiber

structures printed either with the pure PDMS sheath or the composite ink sheath. After curing at 70 °C for 1 h, junction nodes of the 3-layer fiber structure were dissected; morphology of the core-fiber structure at the junction was observed under a microscope (VHX-7000, KEYENCE, Japan).

## Magnetization of the MME structure

n order to design the magnetization profile **m** of the MME structure (see the magnetization process in Supplementary Fig. 5), the geometry of the model was first designed according to the expected deformation of the MME structure driven by the actuation magnetic field. The unmagnetized MME structure was deformed into a preset shape by a mold and magnetized by a pulsed magnetic field $\mathbf{B}_{mag}$ (about 3 T). Under the action of a pulsed magnetic field $\mathbf{B}_{mag}$, the MME structure reaches saturation magnetization, and the magnetization direction is the same as the pulsed magnetic field $\mathbf{B}_{mag}$. When the pulsed magnetic field $\mathbf{B}_{mag}$ disappears, the MME structure forms a stable residual magnetization. Eventually, the designed magnetization profile **m** was formed as the MME structure rebounded to its initial state.

## Morphological characterization of the MME structure

The morphology and corresponding elemental mapping images of MME structures were measured using scanning electron microscopy (Quanta 400 F, FEI, USA) and its energy dispersive spectroscopy (EDS), respectively. NdFeB microparticles and liquid metal distribution inside the MME structure were measured by micro-CT (GE Vtomex). The optical photos of the MME structure were captured by the digital SLR camera (EOS5D, Canon, Japan). The morphology of the MME structure was measured by the ultra-depth three-dimensional microscope (VHX-7000, KEYENCE, Japan).

## Mechanical testing of the MME structure

The coaxially printed fiber was used for tensile experiments (gauge length: 25 mm, outer diameter $d_m$: about 830 μm, inner diameter for the liquid metal core $d_l$: about 270 μm). Nominal stress–stretch curves were plotted for both materials and the specimens were tested on a mechanical testing machine (5967, INSTRON, USA) with a 100 N load cell at a stretching rate of 0.5 mm/s. During the measurement of the fiber cyclic loading-unloading performance, the strain was increased by 30% every 200 cycles until the strain reached 120%.

## Magnetic characterization of the MME structure

The magnetic properties of the ink were evaluated by measuring the magnetic moment density (magnetization) of the composite ink with a vibrating sample magnetometer (7410, Lake Shore, USA). To prepare the specimen, a cube cavity of $2 \times 2 \times 2$ mm$^3$ was prepared by 3D printing to prepare a cube sample suitable for the sample holder of the machine. The magnetic moments of the sample were measured against a sweep of external magnetic fields from −2 T to 2 T. Remnant magnetization was divided by each specimen's volume to obtain the magnetic moment density of the specimen. The remanence of the magnetized MME structure was measured by a Gauss meter (Model 1500, Magnetic Technology Co., Ltd., China).

## Characterization of electrical conductivity

The electrical conductivity of the MME structure was measured by the multimeter (DMM6500, Tektronix Inc, USA). Copper wire electrodes (diameter: 0.25 mm) were attached to the MME structure. For electrical conductivity measurements under cyclic bending, MME coils were printed on polyimide substrates (17 μm in thickness). Cyclic bending of the sample was performed using a custom-made fixture with a controllable bending radius of curvature, stretching an MME fiber with a mechanical testing machine, and measuring the resistance during stretching.

## Characterization of inductance

All the inductance values were measured by a precision inductance, capacitance, and resistance (LCR) meter (Agilent E4980A, Keysight Technologies, USA) at 2 V, 100 kHz. After magnetization, the MME structure was connected to the LCR meter through copper wires, deformed under the action of the magnetic field, and the inductance value was recorded through the LCR meter.

## Fabrication of the MME fiber and coil structure

The MME fiber (Length: 35 mm) and MME coil structure ($35 \times 15 \times 0.8$ mm$^3$) with 8-turn coils were fabricated by coaxial printing. The MME fiber and MME coil structure were magnetized through L-shaped and U-shaped molds, respectively. In the process of magnetically actuated deformation, one end of the MME structure (fiber and coil structure) was fixed. The actuating magnetic field was generated by an electromagnet (WD-80, Changchun Yingpu Magnetic Technology Development Co., Ltd., China.).

## Fabrication of the butterfly robot

The MME coil skeleton was printed directly on the back of the colored paper patagium, which was printed by a traditional color printer (MG3680, Cannon, Japan). After curing, the butterfly robot was cut along the outline of the butterfly robot by laser, and finally, the butterfly robot was obtained (Supplementary Movie 3). Under the constraints of the mold, the wings of this butterfly robot are bent into a flying gesture (the deformation angle is 70°; see Supplementary Fig. 9 for the model and magnetization profile) and magnetized by a 3 T pulsed magnetic field.

## Fabrication of the catheter-style soft surgical tool

The catheter-style soft surgical tool was composed of an MME fiber and a microneedle. The microneedle is installed at the end of the fiber; the other end of the MME fiber was connected to the electrical ablation controller through a conductive wire. The electrical ablation controller (LK-3, Guilin Likang Electronic Medical Equipment Co., Ltd., China) provides the energy of the surgical tool.

## Fabrication of the magnetic navigation system

The magnetic navigation system consists of a current control system, a compensation system, and four electromagnets (diameter: 100 mm, length: 300 mm; see Supplementary Fig. 30 for the magnetic navigation system). The current control system consists of four microcontroller-controlled power amplifier modules that power each electromagnet. The compensation system consists of a stepper motor and a linear module, which controls the movement of the catheter-style soft surgical tool in the $z$ direction. The electromagnets are arranged in orthogonal symmetry.

## Fabrication of the MME gripper

The MME gripper directly printed by coaxial printing was composed of two mutually perpendicular MEE coils (size: $51 \times 9$ mm$^2$). The gripper was bent into a grasping shape under the action of the mold (see Supplementary Fig. 35 for the model and magnetization profile) and magnetized under the pulsed magnetic field (about 3 T). The magnetized MME gripper controls the gripping and release of objects through the magnetic field (the cylindrical NdFeB permanent magnet). At the same time, the inductance of the gripper was measured by the LCR meter. The optical photos and videos were captured by the digital single lens reflex camera (EOS5D, Canon, Japan). A cylindrical NdFeB magnet (diameter: 75 mm and height: 20 mm; N52, Dong Guan City Jun Hui Magnet Co., Ltd., China.) was used to apply magnetic fields required for actuation at a distance. The direction and strength of the applied magnetic fields were varied by manipulating the distance and direction of the magnet through a custom-designed manipulation system (Supplementary Fig. 46). The cylindrical NdFeB permanent

magnet generates the external actuating magnetic field, as shown in Supplementary Fig. 47. To demonstrate the gripping capability and stability of the MME gripper, objects of four different geometries (cube, sphere, cuboid, and convex plate) were 3D printed to have different size and weights. Also, the magnetic field distribution before and after cyclic grasping by the MME gripper (100 cycles; cube: $12 \times 12 \times 12$ mm3; weight: 16.08 g) was measured by a Gauss meter (Model 1500, Magnetic Technology Co., Ltd., China) and the grasping force of the MME gripper before and after cyclic grasping was characterized by a force transducer (407 A, Aurora Scientific, Canada).

### Fabrication of the soft MME robot

The soft MME robot consists of an electromagnetic generator (EMG) module, a radio frequency (RF) wireless energy transmission module and a triboelectric nanogenerator (TENG) module, a rectifier module, and a boost circuit. As for the EMG module, it consists of two oppositely magnetized MME coils (thickness: 0.8 mm, internal dimensions: $3 \times 4$ mm$^2$, external dimensions: $17 \times 18$ mm$^2$, thread pitch: 0.8 mm, and internal resistance: 2.2 Ω), which are directly printed on the flexible circuit board. Place the designed flexible circuit board at the origin of the printing platform. The MME coils were printed directly on the flexible circuit board. Two lead wires were connected to the MME coils to get the electrical output of EMG. After curing, the soft MME robot was placed horizontally and magnetized by a 3 T pulsed magnetic field. The TENG module consists of two copper coil electrodes (thickness: 0.13 mm, Internal dimensions: $3 \times 4$ mm$^2$, external dimensions: $19 \times 20$ mm$^2$, thread pitch: 0.6 mm, and internal resistance: 1.2 Ω), fluorinated ethylene propylene (FEP) film and polyurethane (PU) film. The output open-circuit voltage ($V$) and short-circuit current ($I$) were measured using an electrometer (6514, Keithley, USA) and a multimeter (DMM6500, Tektronix Inc, USA), respectively.

### Finite element analysis

The induced magnetic field, magnetization, and magnetically actuated deformation of coaxial printed MME structures were analyzed using FEA software COMSOL Multiphysics (Version 6.0, COMSOL Inc., Sweden).

### Magnetization of the MME structure by FEA

Magnetization includes two physical fields: magnetic field and mechanical field. So FEA model of the magnetization in the structure was established using a combination of the magnetic field module and the solid mechanics module. The model included an air domain ($500 \times 500 \times 500$ mm$^3$), coil domain (180 mm in diameter), and MME structure (fiber and coil) domain. The geometric meshed with tetrahedral ranging from 0.16 mm to 2.2 mm. The material of the MME structure was set as linear elastic, whose density was 290 kg/m$^3$; the elastic modulus was 870 kPa; and the Poisson's ratio was 0.49. The air domain and the liquid metal materials are set to air and liquid, respectively. The FEA of magnetization includes two analysis steps: the calculation of the MME structure bending process under external force using the solid mechanics module in stationary and the analysis of the MME structure magnetization profile under 3 T pulsed magnetic field using the magnetic field module in a time-dependent. First, the MME structure was bent into a predesigned shape in the solid mechanics module. And then, the folded MME structure was magnetized using a pulse magnetic field (about 3 T) in the magnetism module. The magnetization profile m distributed in the MME structure was calculated.

### Magnetically actuated deformation of the MME structures by FEA

The magnetically actuated deformation of the MME structure also involved multiple physical fields. The FEA model was established by coupling the solid mechanics module with the magnetic field-no current module. The magnetically actuated deformation of each step was analyzed by interactive calculation of magnetic force (calculated by integrating the magnetic stress tensor produced by the actuation magnetic field $\mathbf{B}_{act}$ and magnetization profile $\mathbf{m}$) using the magnetic field-no current module and the deformation using the solid mechanics module. Finally, the final deformation of MME fiber and coil could be obtained by iterative calculation in multi-steps.

### FEA of the butterfly robot, the catheter-style soft surgical tool, the gripper, and the soft robot

The geometric models of the butterfly robot, the catheter-style soft surgical tool, the gripper, and the soft robot were first designed by SolidWorks (Version 2016, Dassault Systemes, France) and imported into COMSOL Multiphysics. These FEA models were meshed with tetrahedral, and their element sizes were set as excessively fined (ranging from 0.24 mm to 9 mm), as shown in Supplementary Fig. 48. The set-ups of the FEA models are the same as these of the FEA model of the MME structure.

### In vivo animal experiments

In vivo experiment was performed with institutional approval from the Animal Care and Use Committee of Sun Yat-Sen University, Guangzhou, China (Approval No. SYSU-IACUC-2022-000244). Male Sprague-Dawley (SD) rats weighing 300–400 g, approximately 8-10 weeks old were used for experiments. In skin and heart ablation, SD rats were anesthetized by intraperitoneal injection (2% pentobarbital, 40 ml/kg). For the experiment of heart surgery, a digital laparoscope (diameter: 0.5 cm) was inserted into the chest cavity via a minimal cut (diameter: about 0.8 cm) in the abdomen. A small incision (diameter: about 0.3 cm) above the target printing location was made to insert the catheter-style soft surgical tool. In the experiment of heart surgery, the image of the rat heart was acquired through a digital laparoscope. Then the coordinates of the ablation points on the $x$-$y$ plane were determined according to the image. The movement of the catheter-style soft surgical tool above the ablation point is controlled by the external actuation magnetic field, and then the compensation system is used to control the catheter-style soft surgical tool to perform compensation movement along the $z$ direction to ensure that the catheter-style soft surgical tool touches the surface of the heart. Furthermore, in order to avoid the interference caused by the beating of the heart, the rat was used for experiments after sacrifice.

### Reporting summary

Further information on research design is available in the Nature Portfolio Reporting Summary linked to this article.

## Data availability

The authors declare that all relevant data supporting the findings of this study are available within the article and its Supplementary Information files. Source data are provided with this paper.

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

## Acknowledgements

This research was supported by the National Natural Science Foundation of China (Project No. 51975597) to L.J., the Natural Science Foundation of Guangdong Province (Project No. 2019A1515011011 and Project No. 2022B1515020011) to L.J.

## Author contributions

Conceptualization: Y.Z., C.P., Z.Li., C.M., L.J. Methodology: Y.Z., C.P., P.L. Experiment: Y.Z., P.L., L.P., Z.Liu, Y.L., Q.W., T.W. Validation: Y.Z., Z.Li., C.M., L.J. Investigation: Y.Z., C.P., Z.Li., C.M., L.J. Writing—original draft: Y.Z., C.P. Writing—review and editing: Z.Li., C.M., L.J. Visualization: Y.Z., C.P., P.L. Supervision: Z.Li., C.M., L.J.

## Competing interests

The authors declare no competing interests.
