## [Peer Review File · Nature Communications]

REVIEWER COMMENTS

Reviewer #1 (Remarks to the Author):

This work demonstrated a hybrid core-sheath fiber composed of a liquid metal (EGaln) core and a soft magnetoactive composite ((NdFeB & PDMS composite) sheath that is patterned using a one-step coaxial printing method. The fibers preserved elastic properties due to fluidic behavior of the metallic core and demonstrated the potential use for biomedical applications, wireless energy transmission/harvesting and magnetic actuation for soft robots. The authors have well characterized experimental results and technical advances of the work. However, the authors may need to emphasize the scientific novelty of the work compared to previous studies utilizing core-sheath fibers with a liquid metal core. The authors may also need to address below to improve the manuscript.

- The manuscript did not discuss enough breakthrough and novelty of the work. Especially, the authors may need to emphasize the novelty of the work in introduction.

- The continuous process to produce the core-sheath fiber with a liquid metal core has been reported recently. The authors may need to also discuss this previous work (Advanced Engineering Materials, 2019, 21, 1900060) in their manuscript.

- Reference 12 might not correctly been cited. Different references might be needed.

- Recently, the liquid metal ferrofluid (i.e., fluidic composites of liquid metal with CIP particles, ACS Applied Materials & Interfaces, 14, 32, 37110) has been utilized instead of magnetization of the elastomers. The authors may be able to demonstrate more dynamic actuation of the fibers if they use the the liquid metal ferrofluid core.

- In Video S1 second demo, is the silicone matrix viscous or dense enough to retard settling of the core liquid metal? Also, towards the end it is seen that the printing is terminated by stacking a layer of core shell fiber over the initial print (14th second onwards). Can the weight of this upper layer lead to collapse of the lower layer or deform the lower structure (since everything is fluid), specially the junctions?

- In Figure S10, it is necessary to provide additional data to explain the led brightness decreasing phenomenon such as reflection coefficient wrt bending angle. Also, can the authors provide theoretical or simulated calculations of the induced voltage in the secondary coil wrt to bending angle? Else, the provided demonstration does not cater any specific scientific information rather than aesthetic appeal.

- For electro mechanical tests of figure 3, it is advisable to conduct a few more tests (a) resistance test by holding under stretched states (viz. 30, 60, 90 and 120 %) to establish that the fiber can maintain constant resistance for long time under stretched conditions and (b) a second round of cyclic tensile test (preferably 100 cycles of 120%) of the same fiber after finishing (a). It is doubtful that if the fiber is kept in a stretched state for long time, micro cracks might be generated if the pdms chains de-graft from the magnetic particles' surfaces. So (b) is necessary.

- For gripping demonstration in figure 6 what is the maximum liftable weight? Under an optimal maximum load, the hands of the gripper should elongate and as such the magnetic properties would change too. It is also advisable to compare magnetic properties of the gripper before and after cyclic loadings to ensure that the properties are retained after fatigue tests.

- How the pdms to magnetic particle ratio of 1:1 was decided?. Was it decided based on optimization of the best rheology suitable for printing purpose (by varying ratios) or was it decided based on increasing magnetic performance?

Reviewer #2 (Remarks to the Author):

Coaxially Printed Magnetic Mechanical Electrical Hybrid Structures with Actuation and Sensing Functionalities

A. Summary of the key results

The paper presents a printable magnetic structure that is capable of sensing and actuation via external magnetic fields. The material exhibits excellent hyperplasticity and electrical conductivity at strains from 30-130% even under harsh environments. The study demonstrates (i) in situ application of electro-ablation surgery, (ii) grasping with the ability to sense grasped object size, and (iii) untethered robot capable of locomotion and wireless energy harvesting.

B. Significance

Typically printed magnetic structures have damaging stress concentrations from internal stiffness mismatch which can lead to delamination by poor interfacial adhesion and material leakage. The study presents a unique solution that utilizes one-step manufacturing to design a core-sheath fiber with liquid metal core and a soft magnetoactive shell. This approach is rapid, untethered, and reversible with added capabilities of remote sensing and energy transmission.

C. Data & methodology

All data and methods are relevant for the study. Nondimensional analysis is used for the guiding of the print to ensure quality. Actuation is characterized independently, and then evaluated in different applications such as in vivo for electro-ablation, and in an untethered robot for locomotion and energy harvesting.

D. Appropriate use of statistics and treatment of uncertainties

Relevant statistical analysis is used for indicating significance. Error bars are used when warranted. Proper model selection when fitting the data.

E. Conclusions

The magnetic-mechanical-electrical core-sheath fiber developed in this study is a catch-all for prominent functionality such as actuation, energy harvesting, and somatosensing. Material and geometry design can be adjusted for desired application (e.g., catheter-style surgery tool). Additionally, hybrid actuation can be achieved in more complex designs as seen in the untethered robot.

F. Suggested improvements

All suggested improvements are included in the clarity section.

G. References

All references are relevant with no key literature missing.

H. Clarity and context

Supplemental videos 5-9 are crowded, and focus appears unclear to the viewer initially.

Small comments include:

Introduction P.1: "Within" is spelled incorrectly.

In Figure 2, is there a reason K is increasing to the left rather than to the right?

Figure 7b is very busy, potentially trying to show too much.

Figure S8 should have LCR specified in the caption.

Video 5: "Stretched" is spelled incorrectly.

Video 9: The "power generation" is unclear on first watch, clearer labeling of what is happening in each phase may be necessary.

Reviewer #3 (Remarks to the Author):

The work by Zhang et al reports a magnetic-mechanical-electrical core-sheath fiber structure based on the combination of liquid metal and soft magnetoactive composite, which has hybrid functions including magnetically actuated deformation, electrical conduction, and so on. The ideas are novel, and the demonstrations are intriguing. The authors have done a solid study of this work from mechanism to application, which has potential to advance the development of liquid metal-based intelligent systems. I recommend accepting the article for publication subject to minor revision.

(1) What are your advantages compared with magnetic liquid metal-soft sheath structure? (Refer to Int. J. Smart Nano Mater. 13 (2), 232-243) Please add a description elaborating on it.

(2) Figure 1(h): If you increase the actuation magnetic field B_{act} , the flapping angle will increase, thus the magnetic flux and its changing rate through the butterfly will decrease. ($B \cos \alpha$) But why does the inductive potential rise instead?

(3) The following articles involve the preparation and application of magnetic liquid metal and liquid metal-based intelligent systems. And I think they may help the authors to improve the article.

1) Sci. adv. 5 (2), eaat4600.

2) Appl. Mater. Today 19, 100597

3) Adv. Mater. 33 (43), 2103062

4) Soft Matter 14 (35), 7113-7118

Response to Reviewers' Comments on "NCOMMS-23-05795"

We would like to thank the reviewers for their constructive remarks and input that help us further improve the manuscript. Each comment from the editor and reviewers has been carefully considered and addressed, with additional experiments. Point-to-point responses to each comment are listed in the following, with corresponding changes highlighted in the revised manuscript in blue for easy tracking.

Reviewer # 1

Comment 1:

However, the authors may need to emphasize the scientific novelty of the work compared to previous studies utilizing core-sheath fibers with a liquid metal core. The manuscript did not discuss enough breakthrough and novelty of the work. Especially, the authors may need to emphasize the novelty of the work in introduction.

Response to Comment 1:

We thank reviewer on this comment. In order to highlight the novelty and breakthrough of this work, we have performed a systematic literature review, revised the Introduction, conducted additional experiments and provided these newly added materials as Supplementary Table 2 and Supplementary Fig. 1 and Fig. 2 in the revised manuscript, which should be enough to highlight the novelty and scientific contribution of our work¹⁻¹⁸.

Through a systematic review, "the liquid-metal sheath structures" reported in the literature can be generally categorized into the 3 different types (see Supplementary Table 2, which is also shown in the following).

As for the Type 1 core-sheath fiber, the core-sheath fibers with a liquid metal core and a polymer sheath do not have magnetoactive properties; soft devices prepared with this structure do not have the hybrid functions of simultaneous magnetic actuation and energy transfer.

As for the Type 2 core-sheath fiber, iron particles or NdFeB particles are added to the liquid metal core, enabling magnetic actuation via the liquid metal ferrofluid. However, the magnetization of the core-sheath fibers of this type (prepared with Fe @ liquid metal ferrofluid) couldn't be customized or

programmed, limiting the magnetic actuation performance¹⁶⁻¹⁷. As for the core-sheath fibers prepared with NdFeB @ liquid metal ferrofluid, it has limited programmable magnetization capability and cannot be programmed into complex geometries (Supplementary Fig. 1c and d).

To deal with these limitations, we presented the Type 3 liquid-metal core-sheath structures in this study (Table S1), consisting of a liquid metal core and a NdFeB @ PDMS sheath. As the NdFeB micro-particles are embedded in the polymer matrix (not in the liquid metal core), this structure has excellent printability, allowing the fabrication of complex 2D or 3D geometries. Moreover, after curing, the magnetic NdFeB micro-particles are immobilized in the polymer sheath, allowing excellent magnetically actuated deformation and complex magnetization programming capabilities (Supplementary Fig. 1)¹⁹⁻²¹. Furthermore, additional experiments demonstrate that this “liquid metal core and NdFeB @ PDMS sheath structure” has nearly the same flexibility as its hollow counterpart without the liquid metal core (Supplementary Fig. 2), demonstrating the excellent flexibility of this structure (in contrast, a high-modulus copper wire would seriously constrain the magnetically induced deformation of fibers²²).

Therefore, as explained in Supplementary Table 2 and Supplementary Fig. 1 and Fig. 2, compared with liquid-metal sheath structures reported in the literature, the liquid-metal sheath structure presented in this study has the following novelty and merits.

(1) From the perspective of material design, a new type of “liquid-metal sheath structure” is presented, enabling the fabrication/printing of complex and customized/programmable 2D/3D geometries with hybrid magnetoactive and electrically conductive characteristics.

(2) From the perspective of device functions, the mechanical-magnetical-electrical properties of the MME structure/device would enable hybrid functions, including programmable magnetization, somatosensory actuation (sensing & wireless actuation) and hybrid/simultaneous actuation & energy transfer, which were not reported according to the best of our knowledge.

In response to this comment, the following revisions have been made in the revised manuscript.

(1) Supplementary Table 2 along with Supplementary Figure 1 and 2 have been added in the revised Supplementary Information to highlight the novelty and breakthrough of this work, which are also shown in the following for your reference.

Supplementary Table 2. The MME fiber in this study in comparison with representative core-sheath fiber structures with a liquid metal core reported in the literature

Different core-sheath structure			Fabrication method	Functional components			Enabled hybrid functions			Typical applications	Ref.
Design	Core material	Sheath material		Mechanical	Electrical	Magnetic	Programming magnetization	Somatosensory actuation	Hybrid actuation & energy transfer		
 LM Liquid metal (LM) PDMS	PU	PDMS	3D shape programming + di-coating	✓	✓	✗	✗	✗	(only wired energy transfer)	 Mechanical sensor Flexible circuit 	[1-2]
	PU	PVDF-HFP-TFE	Coaxial wet-spinning	✓	✓	✗	✗	✗	(only wired energy transfer)	 Pressure sensor Triboelectric Joule heating 	[3-5]
	SEBS	PDMS	Coaxial printing	✓	✓	✗	✗	✗	(both wired and wireless energy transfer)	 Mechanical sensor Pressure sensor Wireless energy transfer 	[6-9]
	Silicone	Ecoflex	SEBS	Template molding + injecting	✓	✓	✗	✗	✗	(only wired energy transfer)	 Metamaterial Stretchable antennas Triboelectric Contactless sensing Phase transition
 LM Fe Magnetic particles & liquid metal PDMS	Ecoflex	Latex		✓	✓	✓	✗	✗	(only wired energy transfer)	 Actuator Variable stiffness 	[17-18]
 LM Liquid metal Magnetic particles (NdFeB) & PDMS PDMS NdFeB			Coaxial printing	✓	✓	✓	✓	✓		 Mechanical sensor Triboelectric Wireless energy transfer Actuator Somatosensory actuation 	Our work

Supplementary Figure 1. Comparison between the MME fiber in this work and existing core-sheath fibers (sheath materials: PDMS or NdFeB @ PDMS; core materials: liquid metal, Fe @ liquid metal or NdFeB @ liquid metal). (a) Schematic illustration of the magnetization process for fibers deformed into different geometries. (b) Experimental results showing magnetically driven deformation of fibers prepared with different materials of different magnetization profiles. After programmed magnetization (L-shape, V-shape, M-shape), our MME fibers can be deformed into the pre-designed shape under magnetic actuation B_{act} . The NdFeB @ liquid metal ferrofluid-based fiber can only be magnetically actuated into simple shapes (cannot be deformed into a complex M-shape). The Fe @ liquid metal ferrofluid-based fiber can only be bended/deformed in a gradient magnetic field B_{gact} , but cannot be deformed into complex shapes. (c) Storage modulus and viscosity of liquid metals with different NdFeB contents. (d) The viscosity of liquid metals with different NdFeB contents. The $B_{act} = 18$ mT, $\nabla|B_{act}| = 0$ mT/mm, and the $B_{gact} = 90$ mT, $\nabla|B_{gact}| = 4$ mT/mm.

Supplementary Figure 2. Comparison of deformation capabilities and flexibility of different core-sheath fibers under magnetic actuation. (a) Deformation of different core-sheath fibers under the same actuation magnetic field ($B_{act} = 18$ mT; core materials: air, copper wire, or liquid metal). (b) Deformation angles for different core-sheath fibers. Unlike the high-modulus copper wire that constrains the deformation, the liquid metal core with high fluidity does not evidently impair the magnetically induced deformation. The MME fiber thus has similar flexibility as its hollow counterpart (filled with air).

(2) Discussion on novelty of this work has been elaborated in “Introduction” of the revised manuscript, which is also shown in the following.

“Previous efforts have explored the fabrication of core-sheath fibers consisting of the liquid metal core and a polymer sheath^{5,23-24}, including template molding and injecting^{9-13,15}, 3D shape programming and dip-coating^{1,18}, coaxial wet-spinning²⁻⁴ and coaxial printing⁵⁻⁷, offering the possibility to fabricate soft electromagnetic devices with complex structures (complex pattern^{8,14} and multilayer structure^{1,6,25}; see Supplementary Table 2 for details). However, these core-sheath fibers only have hybrid mechanical-electrical properties without magnetoactive characteristics, and soft electromagnetic devices built upon these materials lack the hybrid magnetic actuation and energy transfer functions. Although liquid metal ferrofluids²⁶⁻²⁷ could be infused into hollow fibers for magnetically actuated deformation¹⁶⁻¹⁷ while ensuring good flexibility and high electrical conductivity, these liquid metal ferrofluid-based soft electromagnetic devices often suffer from weak remanence or non-programmable magnetization^{19,28-31}, limiting their capabilities for complex shape deformation and somatosensory actuation (see Supplementary Fig. 1 and Fig. 2). Therefore, it is still challenging to develop soft electromagnetic devices with hybrid magnetic actuation, energy transfer and somatosensory actuation functions.” (Line 53 to 66, Page 2, in the revised manuscript)

Comment 2:

The continuous process to produce the core-sheath fiber with a liquid metal core has been reported recently. The authors may need to also discuss this previous work (Advanced Engineering Materials, 2019, 21, 1900060) in their manuscript.

Response to Comment 2:

We thank reviewer for sharing this important work. In this work (*Advanced Engineering Materials, 2019, 21, 1900060*), the authors presented a novel technique to directly print stretchable and flexible electronic devices by coaxially extruding a liquid metal (EGaIn) core with an elastomeric shell (styrene-ethylene-butylene-styrene, SEBS). However, as the magnetic component was not integrated in this material system, the core-sheath fibers presented in this study was not magnetically active and could be used for magnetic actuation.

In repose to this comment, we have discussed this work in the revised manuscript, which is also shown in the following for your reference:

“Previous efforts have explored the fabrication of core-sheath fibers consisting of the liquid metal core and a polymer sheath^{5,23-24}, including template molding and injecting^{9-13,15}, 3D shape programming and dip-coating^{1,18}, coaxial wet-spinning²⁻⁴ and coaxial printing⁵⁻⁷, offering the possibility to fabricate soft electromagnetic devices with complex structures (complex pattern^{8,14} and multilayer structure^{1,6,25}; see Supplementary Table 2 for details). However, these core-sheath fibers only have hybrid mechanical-electrical properties without magnetoactive characteristics, and soft electromagnetic devices built upon these materials lack the hybrid magnetic actuation and energy transfer functions.” (Line 53 to 60, Page 2, in the revised manuscript)

Moreover, we have performed a systematic literature review to compared different core-sheath fibers with a liquid metal core, which has been elaborated in Introduction of the revised manuscript (Page 2), along with Supplementary Table 2 (please refer to Response to Comment 1 from Reviewer 1).

Comment 3:

Reference 12 might not correctly been cited. Different references might be needed.

Response to Comment 3:

Thank you very much for pointing out this typo. In the revised manuscript, in order to highlight the novelty and breakthrough of our work, we have added new references “Ref. 47-70” in Introduction of the revised manuscript. Reference 12 as mentioned by the reviewer has been replaced with other

references in revised manuscript.

Comment 4:

Recently, the liquid metal ferrofluid (i.e., fluidic composites of liquid metal with CIP particles, ACS Applied Materials & Interfaces, 14, 32, 37110) has been utilized instead of magnetization of the elastomers. The authors may be able to demonstrate more dynamic actuation of the fibers if they use the liquid metal ferrofluid core.

Response to Comment 4:

In response to this comment, we have fabricated two fibers with two different liquid metal ferrofluids as reported in the literature (Fe @ liquid metal, following representative studies of “ACS Applied Materials & Interfaces, 14, 32, 37110” and “Int. J. Smart Nano Mater. 13 (2), 232-243”; NdFeB @ liquid metal, following the study of “Matter, 2023, 6(3):855-872”). Performance of these two different fibers with a liquid metal ferrofluid core has been compared with the MME fiber presented in our work. The results are summarized in Supplementary Fig. 1, which is also shown in the following for your reference.

As for the NdFeB @ liquid metal ferrofluid-based fiber, it can be driven by a uniform magnetic field with controllable large-angle deformation. Also, the high viscosity of NdFeB @ liquid metal ferrofluid would partially restrict the free motion of NdFeB particles, enabling programmable magnetization to some extent with low geometry fidelity (see Supplementary Fig. 1b-middle). Also, this fiber could not be programmed to complex geometries such as “M” shape (see Supplementary Fig. 1b-middle right).

As for the Fe @ liquid metal ferrofluid-based fiber, as the iron particles are soft magnetic materials, the Fe @ liquid metal ferrofluid-based fiber can only be slightly deformed by a gradient magnetic field (see Supplementary Fig. 1-bottom). Also, as the iron particles could freely move in the liquid core without much constraint, this fiber could not be magnetized with customization or programming, limiting its magnetic actuation performance and complex shape programming capability.

In contrast, the MME fiber presented this study (with liquid metal core and NdFeB @ PDMS sheath) has excellent magnetically actuated deformation and complex shape programming capabilities.

In response to this comment, the following changes have been made in the revised manuscript:

(1) A new Supplementary Figure 1 has been added in the Supplementary Information (also shown in the following).

(2) “Although liquid metal ferrofluids²⁶⁻²⁷ could be infused into hollow fibers for magnetically

actuated deformation¹⁶⁻¹⁷ while ensuring good flexibility and high electrical conductivity, these liquid metal ferrofluid-based soft electromagnetic devices often suffer from weak remanence or non-programmable magnetization^{19,28-31}, limiting their capabilities for complex shape deformation and somatosensory actuation (see Supplementary Fig. 1 and Fig. 2). Therefore, it is still challenging to develop soft electromagnetic devices with hybrid magnetic actuation, energy transfer and somatosensory actuation functions.” (Line 60 to 66, Page 2, in the revised manuscript)

Supplementary Figure 1. Comparison between the MME fiber in this work and existing core-sheath fibers (sheath materials: PDMS or NdFeB @ PDMS; core materials: liquid metal, Fe @ liquid metal or NdFeB @ liquid metal). (a) Schematic illustration of the magnetization process for fibers deformed into different geometries. (b) Experimental results showing magnetically driven deformation of fibers prepared with different materials of different magnetization profiles. After programmed magnetization (L-shape, V-shape, M-shape), our MME fibers can be deformed into the pre-designed shape under magnetic actuation B_{act} . The NdFeB @ liquid metal ferrofluid-based fiber can only be magnetically actuated into simple shapes (cannot be deformed into a complex M-shape). The Fe @ liquid metal ferrofluid-based fiber can only be bended/deformed in a gradient magnetic field B_{gact} , but cannot be deformed into complex shapes. (c) Storage modulus and viscosity of liquid metals with different NdFeB contents. (d) The viscosity of liquid metals with different NdFeB contents. The $B_{act} = 18$ mT, $\nabla|B_{act}| = 0$ mT/mm, and the $B_{gact} = 90$ mT, $\nabla|B_{gact}| = 4$ mT/mm.

Comment 5:

In Video S1 second demo, is the silicone matrix viscous or dense enough to retard settling of the core liquid metal? Also, towards the end it is seen that the printing is terminated by stacking a layer of core shell fiber over the initial print (14th second onwards). Can the weight of this upper layer lead to collapse of the lower layer or deform the lower structure (since everything is fluid), specially the junctions?

Response to Comment 5:

Thanks for this comment. We compared the printing performance for two materials (pure PDMS: 0 wt% NdFeB in PDMS; the composite ink: 50 wt% NdFeB in PDMS) in the Video S1. We printed a 3-layer fiber structure with these two different sheath materials and then placed an object on the printed structure (mimicking the weights exerted on lower layers by upper layers) in the curing process (Supplementary Fig. 20a).

Although the liquid metal has a density ($\sim 6.5\text{g/cm}^3$) larger than that of the pure PDMS or the composite ink (density: 1g/cm^3 for pure PDMS; $\sim 1.76\text{g/cm}^3$ for the composite ink), as the pure PDMS and the composite ink both have shear thinning properties (Supplementary Fig. 20c), the NdFeB & PDMS composite ink after printing (the shear stress is removed) would exhibit solid-like behavior, which can effectively prevent the settling of the core liquid metal.

In the test mimicking a 30-layer structure, both the pure PDMS and composite ink sheath maintained good structural integrity (Supplementary Fig. 20b). The very bottom layer of the structure was not collapsed by the weight of upper layers, and no settlement of the core liquid metal was observed. Even under the weight of 100 layers, although deformation of the fiber at the bottom layer was observed (which obviously was owing to the exerted weight), an integral core-sheath structure was still maintained and the liquid metal core was not “squeezed” out to rupture the sheath. These results therefore demonstrate the good printability of the MME fiber.

As a further demonstration, we printed two three-layer coil structures with two different inks for the sheath (pure PDMS or the composite ink); the structures were left in the ambient environment (25°C) for 12 hours. After curing in the oven at 70°C , both structures still had good conductivity (Supplementary Fig. 20d). This implies that even left uncured for some time, the composite ink can effectively prevent the settling of liquid metal core.

Therefore, with the composite ink and the co-axial printing method developed in this study, we are able to print 2D/3D MME structures with high geometry fidelity; no settling of the liquid metal core or collapsing of lower layers by upper layers would be observed.

In response to this comment, the following revisions have been made:

- (1) Supplementary Figure 20 has been added in the revised Supplementary Information.
- (2) “Also, after printing (before curing), when the shear stress from printing is released, the composite sheath would exhibit solid-like behavior, which can effectively prevent the settling of the core liquid metal and help maintain the structural fidelity of the printed core-sheath structure. (Supplementary Fig. 20)” (Line 197 to 199, Page 7, in the revised manuscript)
- (3) “In order to demonstrate the stability of the printing performance, a 3-layer fiber structure was

printed with two different sheath materials (pure PDMS: 0 wt% NdFeB in PDMS; the composite ink: 50 wt% NdFeB in PDMS; both inter-fiber spacing and inter-layer spacing are 830 μm). After printing, an object (cubic; size: 10 mm by 10 mm) was placed on top of the printed uncured structure mimicking the weights exerted on bottom layers by the upper layers in the curing process (Supplementary Fig. 20). Weights of the object were adjusted to mimic 30 or 100 layers fiber structures printed either with the pure PDMS sheath or the composite ink sheath. After curing at 70 °C for 1 hour, junction nodes of the 3-layer fiber structure were dissected; morphology of the core-fiber structure at the junction was observed under microscope (VHX-7000, KEYENCE, Japan).” (Line 585 to 594, Page 20, Methods in the revised manuscript)

Supplementary Figure 20. Printing performance with two different sheath materials (pure PDMS: 0 wt% NdFeB in PDMS; the composite ink: 50 wt% NdFeB in PDMS). (a) The shear storage modulus G' and shear loss modulus G'' of composite inks containing different weight fractions of NdFeB particles. (b) Schematic illustration of coaxially printed structures with different equivalent layer thicknesses; weight of the 30-layer structure for the pure PDMS and the composite ink sheath is 1.56 g and 1.83 g respectively; weight of the 100-layer structure for the pure PDMS and the composite ink sheath is 5.2 g and 6.1 g, respectively. (c) Deformation of the fiber at the junction. In the test mimicking a 30-layer structure, both the pure PDMS and composite ink sheath maintained good structural integrity. Under the weight of 100 layers, although deformation of the fiber at the bottom layer was observed (which obviously was owing to the exerted weight), an integral core-sheath structure was still maintained and the liquid metal core was not “squeezed” out to rupture the sheath. (d) Coaxially printed three-layer coil structures left at ambient environments before curing. The three-layer coil structures were printed with two different inks (pure PDMS, or the composite ink). Even left at the ambient environment (25 °C) for 12 hours, the uncured structure (including the connecting fibers in the dangling position at the very top layer) maintained excellent structural fidelity. After curing at 70 °C for 1 hour, the cured three-layer coil structures demonstrated good conductivity and flexibility as their counterparts that were cured immediately after printing.

Comment 6:

In Figure S10, it is necessary to provide additional data to explain the led brightness decreasing phenomenon such as reflection coefficient wrt bending angle. Also, can the authors provide theoretical or simulated calculations of the induced voltage in the secondary coil wrt to bending angle? Else, the provided demonstration does not cater any specific scientific information rather than aesthetic appeal.

Response to Comment 6:

Thank for this comment. In response to this comment, we have performed additional experiments and simulations to 1) explain the led brightness decreasing phenomenon with reference to the bending angle, and 2) provide simulation of the induced voltage in the secondary coil with reference to the bending angle.

As the led brightness (with reference to the bending angle) is related to the induced voltage in the secondary coil, we thus established a finite element analysis model to investigate butterfly robot's radio frequency wireless energy transmission performance and calculate secondary coil's induced voltage under different bending angles.

Supplementary Figure 12 summarizes the simulation results in comparison with the experimental results. As shown in Supplementary Fig. 12b, for the RF wireless energy transmission, strength of the alternating magnetic field (~ 110 kHz) generated by the primary coil would decrease as the distance between the primary coil and the secondary coil increases. As the bending angle α increases, butterfly robot's equivalent area in the x - y plane decreases, thus the magnetic flux passing through the secondary coil embedded within the butterfly robot would decrease (Supplementary Fig. 12d). This would lead to a reduced voltage induced in the secondary coil (Supplementary Fig. 12e). As the LED is lit by the induced voltage generated by the secondary coil, the brightness of the LED would thus decrease with the increase of the deformation angle (Supplementary Fig. 12c).

In response to this comment, the following revisions have been made:

- (1) Supplementary Figure 12 has been revised in the Supplementary Information.

Revised Supplementary Figure 12. Performance of the radio frequency (RF) wireless energy transmission of the butterfly robot. (a) Schematic illustration of the RF wireless energy transmission. The printed 2D MME coil skeleton served as the secondary coil in the butterfly robot for wireless power transmission. (b) Magnetic field distribution of the butterfly robot at different bending angles in the RF wireless energy transmission. (c) Along with the deformation of the butterfly robot (driven by a low-frequency actuation magnetic field B_{act}), LEDs on the butterfly robot can also be lit by another high-frequency magnetic field generated by the RF coil. As the deformation angle α increases, the induced voltage would decrease, and brightness of the LED light decreases correspondingly. (d) The magnetic flux variation of the butterfly robot at different bending angle α . As the bending angle α of the butterfly robot increases, the magnetic flux decreases gradually. (e) Simulation and experimental results of the induced voltage of the secondary coil under different bending angle α . The induced voltage of the secondary coil would decrease with the increase of the bending angle α .

Comment 7:

For electro mechanical tests of figure 3, it is advisable to conduct a few more tests (a) resistance test by holding under stretched states (viz. 30, 60, 90 and 120 %) to establish that the fiber can maintain constant resistance for long time under stretched conditions and (b) a second round of cyclic tensile test (preferably 100 cycles of 120%) of the same fiber after finishing (a). It is doubtful that if the fiber is kept in a stretched state for long time, micro cracks might be generated if the pdms chains de-graft from the magnetic particles' surfaces. So (b) is necessary.

Response to Comment 7:

We thank reviewer for this constructive comment. To clarify reviewer's concern over this issue, we have performed additional experiment, including (a) resistance tests by stretching and holding the

MME fiber at different stretch ratios (strain: 30, 60, 90 and 120 %; maintained for 300 s), (b) cycle test for 100 to 1000 cycles of 60% and 120% stretching, respectively, (c) characterization of surface morphology of the MME fiber to examine any possible micro-cracks.

The results are provided in Supplementary Fig. 22 in the revised Supplementary Information (also shown in the following for your reference).

In the constant strain tests, as the MME fiber was stretched, the sheath thickness or core diameter decreased proportionally (Supplementary Fig. 22b), leading to increased electrical resistance (see Supplementary Fig. 22a1). In this process, due to the high fluidity of the liquid metal, the liquid metal maintained good contact with the composite sheath. Also, owing to the good elasticity, when the stretched MME fiber was maintained at constant strain for 300 seconds, the electrical resistance was stable, with a resistance change $< 1\%$ (see Supplementary Fig. 22a1).

In the cyclic stretching test (tensile strain: 120%), the electrical resistance was stable even after 100 cycles' stretching (Supplementary Fig. 22a2). At a medium strain of 60%, after 1000 cycles' tensile testing, the maximum tensile stress of the MME fiber was quite stable with a change of about 1.8% (Supplementary Fig. 22c and d). Further, at a high strain of 120% (large deformation), the maximum tensile stress of the MME fiber decreased from 1.66 MPa (at the 1st cycle) to 1.46 MPa (at the 1000th cycle), with a reduction of $\sim 12\%$ (Supplementary Fig. 22e and f). This would be owing to the viscoelastic properties of the silicone polymers; similar phenomenon has been reported in the literature³². However, in this process, albeit the large deformation (with reduced maximum tensile stress), the MME fiber maintained its structural integrity; as shown in Supplementary Fig. 22g, no micro-crack could be detected on the outer surface or the cross-section of the MEE fiber after 1000 cycles' testing, demonstrating great mechanical durability of the MME fiber under cyclic loading.

In response to this comment, the following revisions have been made in the revised manuscript:

(1) A new Supplementary Figure 22 has been added in the revised Supplementary Information.

(2) “At a large strain level of 120%, the maximum stress decreased by 12% after 1000 cycles' stretching (Supplementary Fig. 22e and f), which would be owing to the viscoelastic properties of the PDMS substrate³². But, at a moderate deformation (60% strain), the MME fiber showed stable mechanical behaviors (the maximum stress only reduced by 1.8%; Supplementary Fig. 22c and d). Also, after 1000 cycles' fatigue testing, surface morphology of the MME fiber remained unchanged (Supplementary Fig. 22g), no micro-crack could be detected on the outer surface or the cross-section of the MME fiber, demonstrating the great durability of the MME fiber.” (Line 230 to 237, Page 8, in the revised manuscript)

(3) “Moreover, the MME fiber can maintain a constant electrical resistance in the stretched state

(viz. maintained at 30, 60, 90 and 120 % strain for 300 s; resistance change <1%); in the cyclic fatigue test (120% strain), the electrical resistance was also stable (Supplementary Fig. 22a-b).” (Line 251 to 254, Page 9, in the revised manuscript)

Supplementary Figure 22. Characterization of the electrical and mechanical stability and durability of the MME fiber. (a1) Electrical resistance in stretching tests of different tensile strains (fiber length: 35 mm; initial electrical resistance R_0 : $\sim 0.31 \Omega$); the MME fiber was stretched to different strains, maintained at the maximum strain for 300 seconds and then unloaded; loading and unloading rate: 1mm/s. (a2) Changes in electrical resistance during the cyclic stretching (tensile strain: 120%); loading/unloading rate: 1mm/s. (b) Optical images of the pristine core-sheath fiber and the stretched fiber at 120% strain. (c) Cyclic loading-unloading tests for the MME fiber (stretching rate: 1 mm/s; tensile strain: 60%). After 1000 cycles’ tensile fatigue tests, the maximum stress of the fiber was reduced from 0.55 MPa to 0.54 MPa (reduced by 1.8%). (d) Tensile force experienced by the MME fiber during the cyclic loading-unloading tests at a tensile strain of 60%. (e) Cyclic loading-unloading tests for the MME fiber (stretching rate: 1 mm/s; tensile strain: 120%). After 1000 cycles’ tensile tests, the maximum stress of the fiber was reduced from 1.66 MPa to 1.46 MPa (reduced by 12%). (f) Variation of the tensile force experienced by the MME fiber during the cyclic loading-unloading tests at a tensile strain of 120%. (g) Optical and SEM images showing the morphology of the core-sheath MME fiber (before stretching and after 1000 cycles’ stretching). After 1000 cycles’ testing, no obvious micro-crack or particle detachment would be detected on the surface or the cross-section.

Comment 8:

For gripping demonstration in figure 6 what is the maximum liftable weight? Under an optimal

maximum load, the hands of the gripper should elongate and as such the magnetic properties would change too. It is also advisable to compare magnetic properties of the gripper before and after cyclic loadings to ensure that the properties are retained after fatigue tests.

Response to Comment 8:

Thanks for this comment. To address this comment, we have performed additional experiments to (1) characterize the maximum liftable weight of the gripper, and (2) compare magnetic properties of the gripper before and after cyclic loadings. The results are shown in Supplementary Fig. 39 in the revised Supplementary Information.

Regarding the maximum liftable weight of the gripper, it would be affected by both the geometry/shape of the object and the pressure applied onto the object by the gripper (which in turn is affected by the strength of the magnetic field). Thus, the maximum liftable weight has been investigated by lifting objects of different geometries (cube, ball, cuboid, and convex plate) with the MME gripper driven by magnetic field of different strengths (Supplementary Fig. 39a).

As shown in Supplementary Fig. 39b, the maximum lifting weight of the MME gripper increases along with the increase of magnetic field strength and object size. Under magnetic actuation (90 mT), the maximum lifting weights for cubes with different feature lengths (12 mm, 14 mm, 16 mm) are 11.29 g, 14.6 g and 16.08 g, respectively; the maximum lifting weights for balls with different diameters (12 mm, 14 mm, 16 mm) are 22.5 g, 25.7 g and 32.85 g, respectively; for convex plate, the maximum grasping weight is 39.1 g; for cuboid, the maximum grasping weight is 17.67 g.

Regarding the magnetic properties of the MME gripper after fatigue tests, we performed 100 cycles' gripping tests (object: cube: $12 \times 12 \times 12 \text{ mm}^3$, weight: 16.08 g) and characterized the magnetic field distribution and magnetic force of the MME gripper. The results are shown in Supplementary Fig. 39c and d. It is shown that after 100 cycles' gripping tests, the magnetic field distribution of the MME gripper is nearly the same as its initial states (before the 1st cycle of gripping; Supplementary Fig. 39c) and the gripping force is also constant without detectable degradation (Supplementary Fig. 39d). Thus, after cyclic gripping tests, the MME gripper can maintain a stable magnetic property and lifting capabilities.

To deal with comment, the following revisions have been made in the revised manuscript:

(1) A new Supplementary Figure 39 has been added in the revised Supplementary Information.

(2) “The maximum gripping weight would be affected by both the object geometry/shape and the pressure applied onto the object by the gripper (which in turn is affected by the strength of the magnetic field; Supplementary Fig. 39a-b). In addition, after cyclic gripping tests, the MME gripper still had stable magnetic properties and lifting capabilities (Supplementary Fig. 39c-d).” (Line 455 to 459, Page 16, in the revised manuscript)

(3) “To demonstrate the gripping capability and stability of the MME gripper, objects of four different geometries (cube, sphere, cuboid, and convex plate) were 3D printed to have different size and weights. Also, the magnetic field distribution before and after cyclic grasping by the MME gripper (100 cycles; cube: $12 \times 12 \times 12 \text{ mm}^3$; weight: 16.08 g) was measured by a Gauss meter (Model 1500, Magnetic Technology Co., Ltd., China) and the grasping force of the MME gripper before and after cyclic grasping was characterized by a force transducer (407A, Aurora Scientific, Canada).” (Line 672 to 677, Page 22, Methods in the revised manuscript)

Supplementary Figure 39. Gripping capability and durability of the MME gripper. (a) Grasping objects of different shapes by the MME gripper. (b) The maximum weight for objects with different shapes that can be grasped by the MME gripper under different magnetic strength. (c) Experimental results of the magnetic field distribution for the MME gripper before and after cyclic grasping (100 cycles; cube: $12 \times 12 \times 12 \text{ mm}^3$; weight: 16.08 g). (d) The grasping force of the MME gripper before and after cyclic grasping as a function of the strength of the actuation magnetic field.

Comment 9:

How the pdms to magnetic particle ratio of 1:1 was decided?. Was it decided based on optimization of the best rheology suitable for printing purpose (by varying ratios) or was it decided based on increasing magnetic performance?

Response to Comment 9:

We had systematically investigated the effect of NdFeB content on magnetic, rheological, and

mechanical properties of composite ink, leading to the conclusion that 50 wt% was the optimal composition. However, these results were not included in the Supplementary Information in the last submission., Here, we have provided these results in the revised Supplementary Information as Supplementary Fig. 4.

Briefly, we tested the magnetic, rheological and mechanical properties of composite inks with different weight contents of NdFeB particles (0 wt%, 33 wt%, 50 wt%, 67 wt%, and 75 wt%), and systematically characterized the properties of composite inks (remanence, yield stress, tensile strength and elastic modulus) with different NdFeB weight contents (Supplementary Fig. 4). The remanence increased with the content of NdFeB. Thus, a high content of NdFeB can enhance the magnetically actuated deformability of the MME fiber/structure (Supplementary Fig. 4a). However, a higher content of NdFeB also increases the yield stress of the composite ink (Supplementary Fig. 4b), impairing the printability and reducing the flexibility of the printed MME fiber/structure. (Supplementary Fig. 4c and d). Therefore, after comprehensive tests, the composite ink with a NdFeB content of 50wt% (the mass ratio of PDMS to NdFeB particles is 1:1) is selected as the optimal ink for printing the MME structure (Supplementary Fig. 4e).

In response to this comment, the following revisions have been made in the revised manuscript:

(1) The Supplementary Figure 4 has been revised in the Supplementary Information.

(2) “Briefly, a printable composite ink for the sheath was prepared by dispersing non-magnetized NdFeB microparticles (size: 5 \$\mu\$ m) in the PDMS matrix (SE 1700); after systematic investigation of the magnetic, rheological and mechanical properties, the composite ink was optimized to have a NdFeB to PDMS weight ratio of 1:1 (Supplementary Fig. 4).” (Line 92 to 95, Page 3, in the revised manuscript)

Revised Supplementary Figure 4. Properties of composite inks with different NdFeB weight ratios. (a) Magnetization hysteresis loops for composite inks with different NdFeB weight ratios. (b) The shear storage modulus G' and shear loss modulus G'' for different composite inks. (c) The tensile stress-strain curves for fibers printed with composite inks of different NdFeB weight ratios (stretching rate: 1 mm/s). (d) Elastic modulus of fibers printed with composite inks of different NdFeB weight ratios. (e) Heatmap summarizing the properties (remanence, yield stress, tensile strength, and elastic modulus) of composite inks with different weight ratios of NdFeB.

Reviewer #2

Comment 1:

Introduction P.1: “Within” is spelled incorrectly.

Response to Comment 1:

We have carefully checked the manuscript and revised the spelling error in the manuscript.

Comment 2:

In Figure 2, is there a reason K is increasing to the left rather than to the right?

Response to Comment 2:

Thanks for this comment. Previously, a dimensionless parameter K ($K = V/V^*$) was defined to investigate morphology of MME fibers printed under different parameters. In the last version of Fig. 2c, the printing speed was decreasing from the left to the right, thus K would increase to the left as the reviewer commented. To make the information in the figure more intuitive and understandable to readers, we have re-defined K as V^*/V , thus K would increase from the left to the right as the reviewer requested. We have revised the corresponding sections in the manuscript.

In response to this comment, the following revision has been made in the revised manuscript:

“which can be represented by a nondimensional parameter K ($K = V^*/V$). For $K < 1$ (i.e., higher printing speed), the printed MME fiber will be over stretched, leading to a discontinuous fiber, as shown in Fig. 2c-left. A low printing speed (i.e., $K > 1.67$) would lead to insufficient stretching of the extruded uncured fiber, resulting in a serpentine structure with material accumulation (Fig. 2c-right). Thus, appropriate printing speed relative to the feeding rate (i.e., $1 < K < 1.67$) is required to form a continuous core-sheath MME fiber (Fig. 2c-middle).” (Line 171 to 176, Page 6, in the revised manuscript)

Revised Figure 2c. (c) Morphologies of an MME fiber printed under different K ($K = V^*/V$) at a constant V^* of 1.0.

Comment 3:

Figure 7b is very busy, potentially trying to show too much.

Response to Comment 3:

Thanks for this comment. We have revised Fig. 7b (also shown in the following), which would be neat and concise for easy understanding.

Revised Figure 7b. (b) Magnetically controlled rotation and translation of the soft MME robot. Actuated by a rotating magnetic field and a gradient magnetic field, the soft MME robot performs rotational and translational motions (I) and passes through the maze (II). (III) Along with the motion/deformation, the soft MME robot can generate energy in three modes (Mode 1: low-frequency electromagnetic power generation to light up the red LED; Mode 2: high-frequency electromagnetic power generation to light up the red LED; Mode 3: triboelectric power generation to light up the green LED).

Comment 4:

Figure S8 should have LCR specified in the caption.

Response to Comment 4:

We have specified the LCR meter in the caption of Supplementary Figure 10 (Supplementary Figure 8 in the unrevised manuscript).

In response to this comment, the following revision has been made for Figure 10 in the revised Supplementary Information:

“Supplementary Figure 10. (a) The inductance of the 2D MME coil during the deformation of the butterfly robot is measured by the inductance, capacitance, resistance (LCR) meter”.

Comment 5:

Video 5: “Stretched” is spelled incorrectly.

Response to Comment 5: This has been corrected in the revised Video 5.

Comment 6:

Video 9: The “power generation” is unclear on first watch, clearer labeling of what is happening in each phase may be necessary.

Response to Comment 6:

Thanks for this comment. We have re-designed Video 10 (Video 9 in the unrevised manuscript) and added titles to help readers understand the content in Video 10. In addition, we also revised Videos 5-9 to help readers easily understand the content.

Reviewer # 3

Comment 1:

What are your advantages compared with magnetic liquid metal-soft sheath structure? (Refer to Int. J. Smart Nano Mater. 13 (2), 232-243) Please add a description elaborating on it.

Response to Comment 1:

Thanks for this comment. In fact, our work was partially inspired by this article. This study has been added to the reference list in the revised manuscript.

In this important work (*Int. J. Smart Nano Mater. 13 (2), 232-243*), the authors developed a new type of liquid metal magnetoactive slurries (LMMS); based on this material, a variable stiffness wire with excellent electrical conductivity was demonstrated. This wire can switch its stiffness by a magnetic field, realizing functionalities that cannot be achieved with solid smart materials. Although iron particles were added to the liquid metal to enable the magnetic actuation capability via this wire, magnetization of the fiber couldn't be customized or programmed, limiting its magnetic actuation performance.

Therefore, as explained in Supplementary Table 2 and Supplementary Fig. 1 and Fig. 2, compared with liquid-metal sheath structures reported in the literature, the liquid-metal sheath structure presented in this study has the following novelty and merits.

(1) From the perspective of material design, a new type of “liquid-metal sheath structure” is presented, enabling the fabrication/printing of complex and customized/programmable 2D/3D geometries with hybrid magnetoactive and electrically conductive characteristics.

(2) From the perspective of device functions, the mechanical-magnetical-electrical properties of the MME structure/device would enable hybrid functions, including programmable magnetization, somatosensory actuation (sensing & wireless actuation) and hybrid/simultaneous actuation & energy transfer, which were not reported according to the best of our knowledge.

In response to this comment, the following revisions have been made in the revised manuscript.

(1) Supplementary Table 2 along with Supplementary Fig. 1 and Fig. 2 have been added in the revised Supplementary Information to highlight the novelty and breakthrough of this work, which are also shown in the following for your reference.

(2) The article recommended by the reviewer has been added in the revised manuscript as Ref. 59.

“59 Zhou X., Shu J., Jin H., et al. Variable Stiffness Wires Based on Magnetorheological Liquid Metals. *Int. J. Smart Nano Mater.* **13**, 232-243 (2022).” (Page 26, in the revised manuscript)

(3) “Previous efforts have explored the fabrication of core-sheath fibers consisting of the liquid metal core and a polymer sheath^{5,23-24}, including template molding and injecting^{9-13,15}, 3D shape programming and dip-coating^{1,18}, coaxial wet-spinning²⁻⁴ and coaxial printing⁵⁻⁷, offering the possibility to fabricate soft electromagnetic devices with complex structures (complex pattern^{8,14} and multilayer structure^{1,6,25}; see Supplementary Table 2 for details). However, these core-sheath fibers only have hybrid mechanical-electrical properties without magnetoactive characteristics, and soft electromagnetic devices built upon these materials lack the hybrid magnetic actuation and energy transfer functions. Although liquid metal ferrofluids²⁶⁻²⁷ could be infused into hollow fibers for magnetically actuated deformation¹⁶⁻¹⁷ while ensuring good flexibility and high electrical conductivity, these liquid metal ferrofluid-based soft electromagnetic devices often suffer from weak remanence or non-programmable magnetization^{19,28-31}, limiting their capabilities for complex shape deformation and somatosensory actuation (see Supplementary Fig. 1 and Fig. 2). Therefore, it is still challenging to develop soft electromagnetic devices with hybrid magnetic actuation, energy transfer and somatosensory actuation functions.” (Line 53 to 66, Page 2, in the revised manuscript)

Supplementary Table 2. The MME fiber in this study in comparison with representative core-sheath fiber structures with a liquid metal core reported in the literature

Different core-sheath structure			Fabrication method	Functional components			Enabled hybrid functions			Typical applications	Ref.
Design	Core material	Sheath material		Mechanical	Electrical	Magnetic	Programming magnetization	Somatosensory actuation	Hybrid actuation & energy transfer		
	Liquid metal (LM)	PU PDMS	3D shape programming + di-coating	✓	✓	✗	✗	✗	✗ (only wired energy transfer)	 Mechanical sensor Flexible circuit 	[1-2]
		PU PVDF-HFP-TFE	Coaxial wet-spinning	✓	✓	✗	✗	✗	✗ (only wired energy transfer)	 Pressure sensor Triboelectric Joule heating 	[3-5]
		SEBS PDMS	Coaxial printing	✓	✓	✗	✗	✗	✗ (both wired and wireless energy transfer)	 Mechanical sensor Pressure sensor Wireless energy transfer 	[6-9]
		Silicone Ecoflex SEBS	Template molding + injecting	✓	✓	✗	✗	✗	✗ (only wired energy transfer)	 Metamaterial Stretchable antennas Triboelectric Contactless sensing Phase transition 	[10-16]
	Magnetic particles & liquid metal	Ecoflex Latex	Template molding + injecting	✓	✓	✓	✗	✗	✗ (only wired energy transfer)	 Actuator Variable stiffness 	[17-18]
	Liquid metal Magnetic particles (NdFeB) & PDMS	Coaxial printing	Coaxial printing	✓	✓	✓	✓	✓	✓	 Mechanical sensor Triboelectric Wireless energy transfer Actuator Somatosensory actuation 	Our work

Supplementary Figure 1. Comparison between the MME fiber in this work and existing core-sheath fibers (sheath materials: PDMS or NdFeB @ PDMS; core materials: liquid metal, Fe @ liquid metal or NdFeB @ liquid metal). (a) Schematic illustration of the magnetization process for fibers deformed into different geometries. (b) Experimental results showing magnetically driven deformation of fibers prepared with different materials of different magnetization profiles. After programmed magnetization (L-shape, V-shape, M-shape), our MME fibers can be deformed into the pre-designed shape under magnetic actuation B_{act} . The NdFeB @ liquid metal ferrofluid-based fiber can only be magnetically actuated into simple shapes (cannot be deformed into a complex M-shape). The Fe @ liquid metal ferrofluid-based fiber can only be bended/deformed in a gradient magnetic field B_{gact} , but cannot be deformed into complex shapes. (c) Storage modulus and viscosity of liquid metals with different NdFeB contents. (d) The viscosity of liquid metals with different NdFeB contents. The $B_{act} = 18$ mT, $\nabla|B_{act}| = 0$ mT/mm, and the $B_{gact} = 90$ mT, $\nabla|B_{gact}| = 4$ mT/mm.

Supplementary Figure 2. Comparison of deformation capabilities and flexibility of different core-sheath fibers under magnetic actuation. (a) Deformation of different core-sheath fibers under the same actuation magnetic field ($B_{act} = 18$ mT; core materials: air, copper wire, or liquid metal). (b) Deformation angles for different core-sheath fibers. Unlike the high-modulus copper wire that constrains the deformation, the liquid metal core with high fluidity does not evidently impair the magnetically induced deformation. The MME fiber thus has similar flexibility as its hollow counterpart (filled with air).

Comment 2:

Figure 1(h): If you increase the actuation magnetic field B_{act} , the flapping angle will increase, thus the magnetic flux and its changing rate through the butterfly will decrease. ($B\cos\alpha$) But why does the inductive potential rise instead?

Response to Comment 2:

We have performed additional experiment and simulation to answer this comment. The results are provided in Supplementary Fig. 11 (also shown in the following), showing the relationship between the deformation angle α and the induced voltage under different external magnetic field B_{act} .

Deformation of the butterfly robot by an actuation magnetic field is shown in Supplementary Fig. 11a-b. In this test, the butterfly was placed on a flat substrate (Supplementary Fig. 11b); an upward magnetic field would make the wings deform upwards; but when a downward magnetic field was applied, wings of the butterfly robot would be flattened on the substrate and did not deform further downwards (Supplementary Fig. 11c). During the formation of the butterfly wing, as the deformation angle α increased, the equivalent area on the plane perpendicular to the magnetic field decreased (Supplementary Fig. 11d). For instance, as angle α increased from 0° to 68° (angle α was increased by applying a layer magnetic field B_{act}), the maximum rate of change of magnetic flux reduced by 26%; but in this process, the actuation magnetic field B_{act} increased by 3400% for α from 0° (5.5 mT) to 68° (187 mT; see Supplementary Fig. 11e). Owing to the combinative effects from both deformation and B_{act} , the inductive potential of the butterfly robot (absolute value) would increase as the deformation angle α increases (driven by an increasing actuation magnetic strength B_{act} ; see Supplementary Fig. 11f and g).

In response to this comment, the following revisions have been made:

- (1) The Supplementary Figure 11 has been added in the Supplementary Information.

Supplementary Figure 11. Power generation performance of the butterfly robot driven by a low frequency magnetic field. (a) Measurement of the induced voltage generated by the butterfly robot in response to a low-frequency magnetic field. (b) Deformation of the butterfly robot driven by a low-frequency magnetic field. When a downward the magnetic field is applied, the substrate would limit the deformation of the butterfly robot. (c) Deformation angle of the butterfly robot as a function of the actuation magnetic field strength. (d) The equivalent area S of the butterfly robot in the direction of the magnetic field (or the vertical direction in this test) varies with the deformation angle α . As the deformation angle α increases, the equivalent area decreases gradually. (e) A low-frequency magnetic field with increasing strength ($f_{act} = 2$ Hz) for butterfly actuation. (f) Output voltage of the butterfly robot as a function of the actuation magnetic field strength (5.5-187 mT). The output voltage increases with the increase of the magnetic field strength. (g) Dependence of the output voltage E on the magnetic field strength B_{act} .

Comment 3:

The following articles involve the preparation and application of magnetic liquid metal and liquid metal-based intelligent systems. And I think they may help the authors to improve the article.

1) *Sci. adv.* 5 (2), eaat4600.

2) *Appl. Mater. Today* 19, 100597

3) *Adv. Mater.* 33 (43), 2103062

4) *Soft Matter* 14 (35), 7113-7118

Response to Comment 3:

Thanks for sharing these studies. These articles do provide important information on the preparation and application of magnetic liquid metals and liquid metal-based intelligent systems, and are helpful for readers to get a deeper understand of this field.

In response to this comment, the articles recommended by the reviewer have been added in the revised manuscript as Ref. 19-21, and Ref. 47.

- 19 Shu J., Tang S.Y., Feng Z., et al. Unconventional Locomotion of Liquid Metal Droplets Driven by Magnetic Fields. *Soft Matter* **14**, 7113-7118 (2018).
- 20 Li F., Shu J., Zhang L., et al. Liquid Metal Droplet Robot. *Appl. Mater. Today* **19**, 100597 (2020).
- 21 Shu J., Ge D.A., Wang E., et al. A Liquid Metal Artificial Muscle. *Adv. Mater.* **33**, e2103062 (2021).
- 47 Cooper C.B., Joshipura I.D., Parekh D.P., et al. Toughening Stretchable Fibers Via Serial Fracturing of a Metallic Core. *Sci. Adv.* **5**, eaat4600 (2019). (Page 24, 26, in the revised manuscript)

References

- 1 Li G., Zhang M., Liu S., et al. Three-Dimensional Flexible Electronics Using Solidified Liquid Metal with Regulated Plasticity. *Nat. Electron.* **6**, 154-163 (2023).
- 2 Yu X., Fan W., Liu Y., et al. A One-Step Fabricated Sheath-Core Stretchable Fiber Based on Liquid Metal with Superior Electric Conductivity for Wearable Sensors and Heaters. *Adv. Mater. Technol.* **7**, 2101618 (2022).
- 3 Ning C., Wei C., Sheng F., et al. Scalable One-Step Wet-Spinning of Triboelectric Fibers for Large-Area Power and Sensing Textiles. *Nano Res.* **16**, 7518–7526 (2023).
- 4 Zheng L., Zhu M., Wu B., et al. Conductance-Stable Liquid Metal Sheath-Core Microfibers for Stretchy Smart Fabrics and Self-Powered Sensing. *Sci. Adv.* **7**, eabg4041 (2021).
- 5 Khondoker M.A.H., Ostashek A., Sameoto D. Direct 3d Printing of Stretchable Circuits Via Liquid Metal Co-Extrusion within Thermoplastic Filaments. *Adv. Eng. Mater.* **21**, 1900060 (2019).
- 6 Zhou L.Y., Gao Q., Zhan J.F., et al. Three-Dimensional Printed Wearable Sensors with Liquid Metals for Detecting the Pose of Snakelike Soft Robots. *ACS Appl. Mater. Interfaces* **10**, 23208-23217 (2018).
- 7 Wang Y., Wang Z., Wang Z., et al. Multifunctional Electronic Textiles by Direct 3d Printing of Stretchable Conductive Fibers. *Adv. Electron. Mater.* **9**, 2201194 (2023).
- 8 Chen Y., Liu Y., Ren J., et al. Conformable Core-Shell Fiber Tactile Sensor by Continuous Tubular Deposition Modeling with Water-Based Sacrificial Coaxial Writing. *Mater. Des.* **190**, 108567 (2020).
- 9 Ning C., Dong K., Gao W., et al. Dual-Mode Thermal-Regulating and Self-Powered Pressure Sensing Hybrid Smart Fibers. *Chem. Eng. J.* **420**, 129650 (2021).
- 10 Lai Y.C., Lu H.W., Wu H.M., et al. Elastic Multifunctional Liquid–Metal Fibers for Harvesting Mechanical and Electromagnetic Energy and as Self-Powered Sensors. *Adv. Energy Mater.* **11**, 2100411 (2021).

- 11 Fu C., Tang W., Miao Y., et al. Large-Scalable Fabrication of Liquid Metal-Based Double Helix Core-Spun Yarns for Capacitive Sensing, Energy Harvesting, and Thermal Management. *Nano Energy* **106**, 108078 (2023).
- 12 Cooper C.B., Joshipura I.D., Parekh D.P., et al. Toughening Stretchable Fibers Via Serial Fracturing of a Metallic Core. *Sci. Adv.* **5**, eaat4600 (2019).
- 13 Yamagishi K., Zhou W., Ching T., et al. Ultra-Deformable and Tissue-Adhesive Liquid Metal Antennas with High Wireless Powering Efficiency. *Adv. Mater.* **33**, 2008062 (2021).
- 14 Lin R., Kim H.J., Achavananthadith S., et al. Digitally-Embroidered Liquid Metal Electronic Textiles for Wearable Wireless Systems. *Nat. Commun.* **13**, 2190 (2022).
- 15 Sun X., Fu J.H., Zhao H., et al. Electronic Whiskers for Velocity Sensing Based on the Liquid Metal Hysteresis Effect. *Soft Matter* **18**, 9153-9162 (2022).
- 16 Hong K., Choe M., Kim S., et al. An Ultrastretchable Electrical Switch Fiber with a Magnetic Liquid Metal Core for Remote Magnetic Actuation. *Polymers* **13**, 2407 (2021).
- 17 Zhou X., Shu J., Jin H., et al. Variable Stiffness Wires Based on Magnetorheological Liquid Metals. *Int. J. Smart Nano Mater.* **13**, 232-243 (2022).
- 18 Ma B., Zhang J., Chen G.S., et al. Shape-Programmable Liquid Metal Fibers. *Biosens.-Basel* **13**, 28 (2022).
- 19 Kim Y., Parada G.A., Liu S.D., et al. Ferromagnetic Soft Continuum Robots. *Sci. Rob.* **4**, eaax7329 (2019).
- 20 Hu W., Lum G.Z., Mastrangeli M., et al. Small-Scale Soft-Bodied Robot with Multimodal Locomotion. *Nature* **554**, 81-85 (2018).
- 21 Kim Y., Yuk H., Zhao R., et al. Printing Ferromagnetic Domains for Untethered Fast-Transforming Soft Materials. *Nature* **558**, 274-279 (2018).
- 22 Du Z., Ai J., Zhang X., et al. Stretchable Electromagnetic Fibers for Self-Powered Mechanical Sensing. *Appl. Mater. Today* **20**, 100623 (2020).
- 23 Qi X., Zhao H., Wang L., et al. Underwater Sensing and Warming E-Textiles with Reversible Liquid Metal Electronics. *Chem. Eng. J.* **437**, 135382 (2022).
- 24 Leber A., Dong C., Chandran R., et al. Soft and Stretchable Liquid Metal Transmission Lines as Distributed Probes of Multimodal Deformations. *Nat. Electron.* **3**, 316-326 (2020).
- 25 Wu Q., Zhu F., Wu Z., et al. Suspension Printing of Liquid Metal in Yield-Stress Fluid for Resilient 3d Constructs with Electromagnetic Functions. *npj Flexible Electron.* **6**, 50 (2022).
- 26 Kim S., Kim S., Hong K., et al. Liquid-Metal-Coated Magnetic Particles toward Writable, Nonwetable, Stretchable Circuit Boards, and Directly Assembled Liquid Metal-Elastomer Conductors. *ACS Appl. Mater. Interfaces* **14**, 37110-37119 (2022).
- 27 Wang Q., Pan C., Zhang Y., et al. Magnetoactive Liquid-Solid Phase Transitional Matter. *Matter* **6**, 855-872 (2023).
- 28 Xu T.Q., Zhang J.C., Salehizadeh M., et al. Millimeter-Scale Flexible Robots with Programmable Three-Dimensional

Magnetization and Motions. *Sci. Rob.* **4**, eaav4494 (2019).

- 29 Lin C., Lv J.X., Li Y.S., et al. 4d-Printed Biodegradable and Remotely Controllable Shape Memory Occlusion Devices. *Adv. Funct. Mater.* **29**, 1906569 (2019).
- 30 Cao X., Xuan S., Sun S., et al. 3d Printing Magnetic Actuators for Biomimetic Applications. *ACS Appl. Mater. Interfaces* **13**, 30127-30136 (2021).
- 31 Zhang Y.X., Wang Q.Y., Yi S.Z., et al. 4d Printing of Magnetoactive Soft Materials for on-Demand Magnetic Actuation Transformation. *ACS Appl. Mater. Interfaces* **13**, 4174-4184 (2021).
- 32 Wang Z., Xia X., Zhu M., et al. Rational Assembly of Liquid Metal/Elastomer Lattice Conductors for High-Performance and Strain-Invariant Stretchable Electronics. *Adv. Funct. Mater.* **32**, 2108336 (2021).

REVIEWERS' COMMENTS

Reviewer #1 (Remarks to the Author):

The authors have addressed well the comments raised by the reviewers. I would like to recommend the publication of this revised manuscript.

Reviewer #2 (Remarks to the Author):

The authors have effectively addressed my comments, and I have no additional feedback. I recommend this article for publication.

Reviewer #3 (Remarks to the Author):

The responses to the reviewer comments are sufficient to recommend the paper for publication in this journal.

Response to Reviewers' Comments on "NCOMMS-23-05795A"

Reviewer # 1

Comment 1: *The authors have addressed well the comments raised by the reviewers. I would like to recommend the publication of this revised manuscript.*

Response to Comment 1: We thank this reviewer for the effort on evaluating our work, the highly positive comment, and the great help in the improvement of the manuscript.

Reviewer # 2

Comment 1: *The authors have effectively addressed my comments, and I have no additional feedback. I recommend this article for publication.*

Response to Comment 1: We thank this reviewer for the effort on evaluating our work, the highly positive comment, and the great help in the improvement of the manuscript.

Reviewer #3

Comment 1: *The responses to the reviewer comments are sufficient to recommend the paper for publication in this journal.*

Response to Comment 1: We thank this reviewer for the effort on evaluating our work, the highly positive comment, and the great help in the improvement of the manuscript.